# State-dependent geometry of population activity in rat auditory cortex

**Dmitry Kobak[1,2†], Jose L Pardo-Vazquez[1,3†]\*, Mafalda Valente[1], Christian K Machens[1], Alfonso Renart[1]\***

[1]Champalimaud Center for the Unknown, Lisbon, Portugal; [2]Institute for Ophthalmic Research, University of Tübingen, Tübingen, Germany; [3]Neuroscience and Motor Control Group, University of A Coruña, Coruña, Spain

**Abstract** The accuracy of the neural code depends on the relative embedding of signal and noise in the activity of neural populations. Despite a wealth of theoretical work on population codes, there are few empirical characterizations of the high-dimensional signal and noise subspaces. We studied the geometry of population codes in the rat auditory cortex across brain states along the activation-inactivation continuum, using sounds varying in difference and mean level across the ears. As the cortex becomes more activated, single-hemisphere populations go from preferring contralateral loud sounds to a symmetric preference across lateralizations and intensities, gain-modulation effectively disappears, and the signal and noise subspaces become approximately orthogonal to each other and to the direction corresponding to global activity modulations. Level-invariant decoding of sound lateralization also becomes possible in the active state. Our results provide an empirical foundation for the geometry and state-dependence of cortical population codes.
DOI: https://doi.org/10.7554/eLife.44526.001

**\*For correspondence:**
jose.pardovazquez@neuro.
fchampalimaud.org (JLP-V);
alfonso.renart@neuro.
fchampalimaud.org (AR)

†These authors contributed
equally to this work

**Competing interests:** The
authors declare that no
competing interests exist.

**Reviewing editor:** Emilio
Salinas, Wake Forest School of
Medicine, United States

## Introduction

Sensory stimuli are represented in the brain through the activity of large populations of neurons. While this fact has motivated the study of neural population codes for several decades, the bulk of this work has been theoretical, since only recently has it become feasible to record the simultaneous activity of large neuronal ensembles.

Early theoretical studies relied on parametric models of neuronal firing to quantify how the accuracy of a population code depended on factors such as the shape of neural tuning curves, the dimensionality of sensory stimuli or the duration of spike-count windows (*Seung and Sompolinsky, 1993*; *Zhang and Sejnowski, 1999*; *Bethge et al., 2002*). It was recognized that pairwise 'noise' correlations between the activity of different neurons had a potentially large impact on the accuracy of a population code, and that the effect of correlated variability depends on the relative orientation of the subspaces where the signal and the noise reside (*Abbott and Dayan, 1999*; *Panzeri et al., 1999*; *Sompolinsky et al., 2001*; *Wu et al., 2001*; *Shamir and Sompolinsky, 2006*; *Averbeck et al., 2006*; *Cohen and Kohn, 2011*; *Ecker et al., 2011*; *Moreno-Bote et al., 2014*). The signal subspace describes the set of trial-averaged network states visited by the population as the sensory stimulus is varied. The noise subspace describes the set of states visited by the population as a result of trial-to-trial variability for any given fixed stimulus. If the signal and noise subspaces are aligned, then a putative decoder will be unable to tease apart the stimulus from the noise, and this will decrease the accuracy of the code. Despite the amount of theoretical work on this topic, the geometry of the signal and noise subspaces in large populations of neurons has not yet been thoroughly characterized.

Here, we conducted such an investigation with an emphasis on understanding how the geometry of the population is affected by the global dynamics of the brain. A salient feature of brain dynamics that becomes immediately apparent when global activity is measured — either in the form of populations of single neurons or in the form of mesoscopic signals such as the LFP or EEG — is the existence of different 'global dynamical regimes', or 'brain states' (*Vanderwolf, 2003*; *Harris and Thiele, 2011*; *McCormick et al., 2015*). Behavioral context, such as overt motion during wakefulness (*Vanderwolf, 1988*; *Castro-Alamancos, 2004*; *Gervasoni et al., 2004*), whisking (*Poulet and Petersen, 2008*; *Fanselow and Nicolelis, 1999*), sleep phase (*Steriade et al., 1990*; *Steriade and McCarley, 2013*), or the type of actions that animals are performing (*Vanderwolf, 2003*; *Gervasoni et al., 2004*), have a large impact on brain state. Different brain states are also associated to different neuromodulatory systems (*Vanderwolf, 2003*; *Lydic and Baghdoyan, 1998*; *Lee and Dan, 2012*). From a physiological perspective, different brain states can be arranged along a one-dimensional continuum of cortical activation (*Berger, 1929*; *Harris and Thiele, 2011*). At one end of the continuum (inactive or synchronized state) the population undergoes global, large-amplitude, low-frequency oscillations leading to alternations between periods of firing and of silence referred to as up and down states (*Steriade et al., 1993*). These states are typical of slow-wave sleep, the anesthetized brain under most, but not all, anesthetics, and quiet wakefulness (*Harris and Thiele, 2011*). At the other end of the continuum (active or desynchronized state), firing rates are tonic, and the population-averaged activity of the population shows much weaker fluctuations (*Destexhe et al., 2003*; *Steriade and McCarley, 2013*; *Renart et al., 2010*). Active states are typical of REM sleep, can be observed under urethane anesthesia (*Clement et al., 2008*) and are associated with locomotion and active sampling of the environment, including attentive wakefulness (*Vanderwolf, 2003*; *Harris and Thiele, 2011*). The population can find itself also in intermediate activation states (*Curto et al., 2009*). Since the magnitude and nature of correlated variability change along the activation continuum, brain state can be expected to have a strong influence on the geometry of sensory population codes. Recent studies generally tend to find that the amount of information about the stimulus in a population code is larger during active than during inactive states (*Goard and Dan, 2009*; *Marguet and Harris, 2011*; *Pachitariu et al., 2015*; *Beaman et al., 2017*).

We have studied the state-dependence of the geometry of population codes in the rat auditory cortex. Different brain states were evoked through the use of urethane anesthesia (*Murakami et al., 2005*; *Clement et al., 2008*; *Renart et al., 2010*). We recorded population activity in response to binaural noise bursts varying both in inter-aural level difference (ILD: the difference in sound intensity in dB between the two ears) as well as in absolute binaural level (ABL: the arithmetic average of the left and right intensities). Since rodents do not use phase information for sound localization, they rely mainly on ILD to localize sounds in the horizontal plane (*Wesolek et al., 2010*; *Lauer et al., 2011*). Neurons in the auditory cortex in several species, including rats, represent ILD in the form of broad tuning curves with maximum slope at near-zero ILDs, that is corresponding to sounds arriving from the front of the animal (*Stecker et al., 2005*; *Campbell et al., 2006*; *Yao et al., 2013*). ABL and ILD are prototypical examples of prothetic and metathetic sensory continua, respectively (*Stevens, 1957*). Prothetic sensory continua are associated with changes in the total energy carried by the stimulus, whereas along metathetic continua the total amount of energy stays constant and the stimuli differ along some other feature dimensions — in our case in lateralization. Other examples of metathetic continua include sound frequency, or spatial location and stimulus orientation in vision. By systematically varying ABL and ILD, our study allowed us to investigate how these two types of sensory continua are represented at the population level. The state-dependence of the representation of prothetic stimuli is particularly interesting, both because it has not been systematically explored previously, and because global up-down fluctuations in the inactive state are expected to specially interfere with the representation of stimuli varying along a prothetic continuum (ABL).

## Results

The simultaneous activity of many well-isolated single units was recorded unilaterally from the auditory cortex of urethane-anesthetized rats using 8-shank, 64-channel silicon probes (Neuronexus Tech.). We obtained data from $n = 23$ sessions (after quality control, see Materials and methods), with $104 \pm 23$ isolated neurons per session (mean $\pm$ standard deviation across sessions).

Under urethane anaesthesia, the cortex undergoes spontaneous transitions between states with varying degrees of 'activation', also known as 'desynchronization' (*Clement et al., 2008*). Characteristic examples of these states are shown in *Figure 1* as raster plots (sorted by firing rate), taken from periods of spontaneous activity. As shown previously (*Curto et al., 2009*; *Renart et al., 2010*), during the inactive state, there are globally correlated low-frequency fluctuations in population activity (*Figure 1A*; also known as the 'slow oscillation'). In this state, there are brief periods when all neurons are silent (down states) separated by periods of firing (up states). These up-down oscillations induce positive correlations between most pairs of neurons evident as vertical stripes in the population raster (*Renart et al., 2010*; *Ecker et al., 2014*). In contrast, during the active state, neurons fire tonically in a globally uncorrelated fashion (*Figure 1B*), as evident from the lack of vertical stripes in the raster (*Renart et al., 2010*; *Ecker et al., 2010*).

Our set of auditory stimuli was designed to probe the representation of ILD (the cue used by rodents to localize sound on the horizontal plane [*Wesolek et al., 2010*; *Lauer et al., 2011*]) and its intensity dependence. We varied the ILD of the sounds at constant absolute binaural level (ABL, the arithmetic mean of the level (in dB SPL) across the two ears). We used a headphone-like configuration with two speakers positioned closely (0.5 cm) to each ear in order to replicate the conditions of a behavioral study in our laboratory where we have thoroughly characterized the ILD-discrimination abilities of freely moving rats using custom made headphones (*Pardo-Vazquez et al., 2018*). Since that study focused on ILD discrimination with respect to the midline, our stimulus set also focused on small-magnitude ILDs (±1.5, ±3, ±4.5, ±6, ±12, ±20 dB), which were presented at three different overall intensities (ABL = 20, 40, 60 dB SPL; see *Figure 1C*). Sounds consisted of wide-band (5–20 KHz) noise bursts lasting 150 ms separated by ~850 ms silence. Each of the 36 stimuli was presented 100 times.

We quantified the degree of (in)activation in two different ways. First, we computed the coefficient of variation (CV, standard deviation divided by the mean) of the total number of spikes from all single neurons across 20 ms bins (see Materials and methods). This number — the CV of the population rate — varied from around 0.4 to around 2.2. If all neurons are Poisson and statistically independent, then the CV of the population rate will approach zero. Second, we also quantified the degree of inactivation by the fraction of 20 ms windows corresponding to down states, which varied from 0% to over 70% across sessions. Both measures turned out to be strongly correlated ($r = 0.98$, *Figure 1D*), and we will subsequently use the CV of the population rate to quantify the degree of

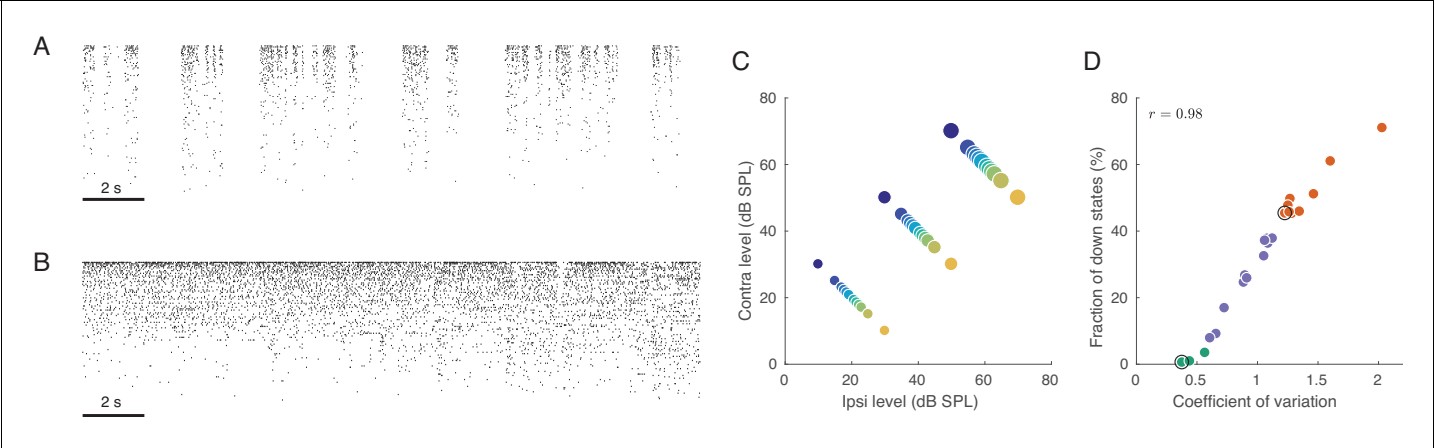

**Figure 1.** Population activity in rat auditory cortex during inactive and active states. (A) Exemplary raster plot of 20 s of spontaneous activity in an inactive state ($N = 141$ neurons, sorted by firing rate). (B) Exemplary raster plot of 20 s of spontaneous activity in an active state ($N = 114$ neurons, sorted by firing rate). (C) Stimulus set. The auditory stimuli used throughout this study consisted of brief broad-band (5–20 KHz) noise bursts. The colors represent ILD and the dot sizes represent ABL. This color/size code is maintained throughout our study. (D) Relationship between coefficient of variation (CV) of the total spike count and the fraction of down states across all recorded sessions. Green/orange dots correspond to sessions that we will use as 'active sessions' and 'inactive sessions' (as the two extremes of the activation-inactivation continuum) but state-dependence will always be assessed using all recordings by regressing various quantities of interest against CV. Black circles show two sessions used for raster plots in (A), (B).
DOI: https://doi.org/10.7554/eLife.44526.002

activation. The distribution of either of these measures is unimodal, suggesting that active and inactive states are not clearly distinct bur rather form a continuum (*Curto et al., 2009*). Thus, we quantify the effect of brain state on different aspects of neural activity using regressions against the value of the CV across sessions. In order to illustrate the state-dependence of various features of interest, we arbitrarily defined all sessions with CV<0.6 as 'active' ($n = 3$) and all sessions with CV>1.2 as 'inactive' ($n = 9$), see *Figure 1D*. This was used only for illustration purposes. In addition, throughout the paper, we will use one example active and one example inactive sessions for illustration purposes.

## Mean evoked activity

We begin with analyzing evoked activity averaged across all neurons within a session. *Figure 2A* shows mean evoked activity for an exemplary inactive session, and *Figure 2B* shows evoked responses additionally averaged across all inactive sessions. We see that the magnitude of the neural response strongly depends on the presented stimulus: loud stimuli evoke stronger responses than silent stimuli, and contralateral stimuli evoke stronger responses than ipsilateral stimuli. The same responses averaged over the whole 150 ms of stimulus duration are shown in *Figure 2C*. This 'tuning curve' clearly shows contralateral as well as loud preference. The same effect is observed in the 'onset' response (20–40 ms, small dots), perhaps even stronger. We observed that an increase in

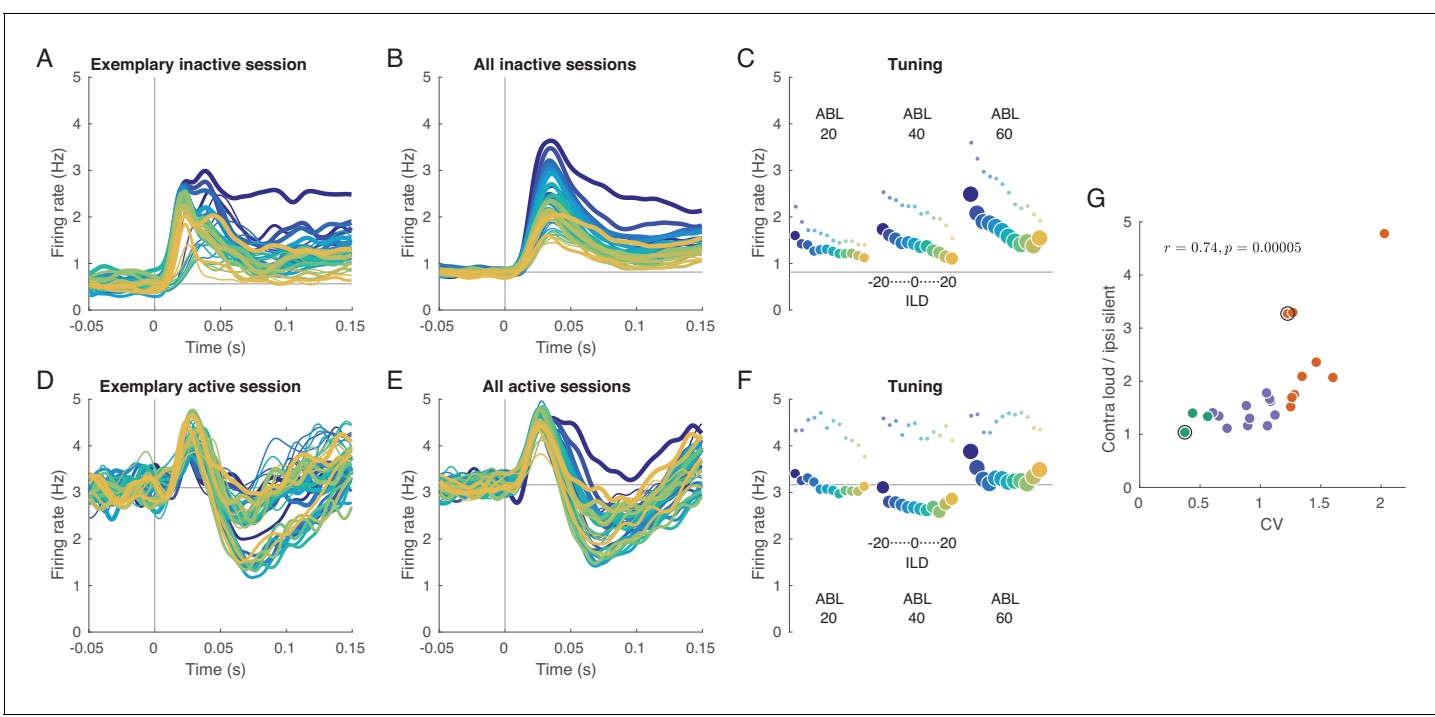

**Figure 2.** Evoked activity in inactive and active states, averaged across neurons. (**A**) Exemplary inactive session. Each line is a population PSTH (peristimulus time histogram) corresponding to one stimulus (color coded as in *Figure 1*, line thickness corresponds to ABL). Population PSTHs were computed by averaging the Gaussian-smoothed ($\sigma = 5$ ms) PSTHs of $N = 141$ single neurons. (**B**) Mean of the population PSTHs across nine inactive sessions. (**C**) Average firing rate over 150 ms of stimulus duration for each stimulus, corresponding to panel B. Dots show the early response, defined as the average over 20–40 ms window. (**D–F**) The same for one exemplary and three overall active sessions. (**G**) The ratio of the evoked responses for the loudest, most contralateral stimuli and the most silent, most ipsilateral stimuli, computed for each session. Green dots are active sessions, orange dots are inactive sessions, black circles mark the two exemplary sessions used in (**A**) and (**D**).
DOI: https://doi.org/10.7554/eLife.44526.003

The following figure supplements are available for figure 2:

**Figure supplement 1.** Evoked responses in the inactive sessions conditioned on the state before the stimulus presentation.
DOI: https://doi.org/10.7554/eLife.44526.004

**Figure supplement 2.** Tuning curves for 50 neurons with the highest average evoked responses in the exemplary inactive (top) and active (bottom) sessions.
DOI: https://doi.org/10.7554/eLife.44526.005

overall stimulus intensity (ABL) leads to the increased slope of the ILD tuning: for ABLs 20, 40, and 60 dB, the absolute value of the slope of firing rate regressed on ILD increased from 0.012 to 0.018 to 0.028 (and from 0.022 to 0.026 to 0.048 for onset responses). This effect of overall stimulus intensity (ABL) as a modulation of the gain of the tuning to ILD is reminiscent of the effect of visual contrast on, for example orientation tuning (*Skottun et al., 1987*). Note, however, that the effect shown in *Figure 2C* is at the level of the population-averaged tuning curve. We address gain modulation at the level of single neurons below.

In the active state the picture is different (*Figure 2D–F*). Here, the responses are similar and strongly overlap without a pronounced monotonic ILD or ABL tuning. Especially close to the midline, which is the most critical stimulus region where behavioral accuracy is the highest (*Heffner and Heffner, 1988*; *Recanzone et al., 1998*; *Recanzone and Beckerman, 2004*), the tuning curve displays very weak contralateral tuning and is approximately flat, meaning that all stimuli evoke roughly the same neural response (number of spikes) in the auditory cortex of a single hemisphere.

To quantify this effect on a session-by-session basis, we computed the ratio of the mean response (mean across neurons and across 150 ms) evoked by the loudest, most contralateral stimulus (large blue circle in *Figure 2C,F*) and that evoked by the faintest, most ipsilateral stimulus (small orange circle). This ratio was positively correlated with CV with $r = 0.74$ (*Figure 2G*, $p = 0.00005$), being close to one in the most active sessions and reaching almost five in the most inactive one.

In the above analysis, we averaged the responses over all trials with the same stimulus. However, in the inactive state any given stimulus will sometimes occur during a down state and sometimes during an up state. Conditioning the analysis on these two situations showed that responses to stimuli arriving during down-states are more similar to the unconditioned responses shown above. However, the responses to stimuli arriving during up-states are still different from those in the active state, particularly in the early onset response. While onset responses in the active state show no clear ILD tuning (*Figure 2F*), onset responses to sounds arriving during up-states show a clear preference for contralateral ILDs (*Figure 2—figure supplement 1*). Given that the difference between responses in the active and inactive states persists when only responses during up-states are considered (see also *Pachitariu et al., 2015*), we will not be conditioning on the phase of the slow oscillation in subsequent analyses.

## Evoked activity of single neurons

The small and unspecific population-averaged responses in the active state could result from a generic lack of tuning of single neurons. Alternatively, single neurons during the active state could be strongly tuned, but with heterogeneous preferences, leading to unspecific population-averaged responses. To distinguish between these two possibilities, we examined the tuning of single cells to ILD and ABL.

As typical in the cortex, the tuning of single neurons was highly diverse. A snapshot of this diversity is shown in *Figure 2—figure supplement 2*, where we plot the tuning curves for the 50 neurons with highest activity in our exemplary recordings of active and inactive states. To summarize the tuning of each single neuron, we regressed its mean firing rate in each condition (computed in the 150 ms window of stimulus presentation and averaged across trials) on the ABL and ILD values. For each neuron, mean firing rates were $z$-scored before the regression, so that the regression coefficients were not affected by high/low overall firing rate and only quantified the strength of linear tuning (see *Figure 3—figure supplement 1* for the same analysis without $z$-scoring). ABL values (possible values: 20, 40, and 60) were centered by subtracting 40. For each neuron, the model takes the following form:

$$z-\text{score}(\text{Mean firing rate}) = \beta_0 + \beta_{\text{ILD}} \cdot \text{ILD} + \beta_{\text{ABL}} \cdot (\text{ABL} - 40) + \beta_{\text{inter}} \cdot (\text{ABL} - 40) \cdot \text{ILD} + \varepsilon. \quad (1)$$

Approximately linear ILD tuning near the midline is expected given previous characterizations (*Stecker et al., 2005*; *Yao et al., 2013*). The interaction term can account for the effects of gain modulation that we described above at the level of the population-averaged response. Note that both ILD and ABL (after subtracting 40) are centered and the $3 \times 12$ design is balanced; it follows that all three linear terms in this model are estimated independently (i.e. each of them corresponds to the value that one would obtain using uni-variate linear regression), and the presence of the interaction term does not influence $\beta_{\text{ILD}}$ and $\beta_{\text{ABL}}$.

Of course, this simple model cannot fully represent the complexity of neural tuning, but it provides a first approximation sufficient to reveal strong differences in tuning between brain states. The quality of the model fit was roughly the same in the inactive and the active states ($R^2 = 0.36 \pm 0.24$ and $0.34 \pm 0.22$ correspondingly, mean $\pm$ SD across all neurons in all active and all inactive sessions). In both states, around one half of the neurons showed significant linear tuning with overall $p<0.01$ (53.5%, 489/914, in the inactive sessions and 49.5%, 154/311, in the active sessions).

We first focused on the way neurons are separately tuned to ILD and ABL, by examining the distribution of coefficients $\beta_{\mathrm{ILD}}$ and $\beta_{\mathrm{ABL}}$ across the population. *Figure 3A* shows these coefficients for each neuron in an exemplary inactive session, with $\beta_{\mathrm{ILD}}$ on the horizontal and $\beta_{\mathrm{ABL}}$ on the vertical axis. Only neurons with significant ($p<0.01$) tuning are shown here; *Figure 3—figure supplement 2* presents the same analysis with all neurons. Most neurons are located in the upper-left quadrant with negative $\beta_{\mathrm{ILD}}$ and positive $\beta_{\mathrm{ABL}}$, meaning that they are tuned to the loud contralateral sounds. *Figure 3B* pools neurons from nine inactive sessions and *Figure 3C* shows the corresponding angular histogram, both supporting the same conclusion. This is consistent with previous work showing a preference for contralateral sounds in the auditory cortex (*Stecker et al., 2005*; *Campbell et al., 2006*; *Yao et al., 2013*).

In the active state the situation is again different (*Figure 3D–F*). The distribution of linear coefficients is roughly symmetric with respect to the horizontal and the vertical axes. As we go around the clock in *Figure 3D*, we see neurons preferring loud, then ipsilateral, then faint, then contralateral sounds (insets show tuning curves of eight exemplary neurons using the same format as in *Figure 2C,F*). Separate analysis of the onset (0–50 ms) and the late (100–150 ms) responses revealed that this strong tuning heterogeneity was already present in the onset responses (*Figure 3—figure supplement 3*), even though there was some contralateral and loud bias in the early responses. Interestingly, in the active state more neurons were significantly tuned in the late response compared to the early response (41.8% vs. 23.1%, *Figure 3—figure supplement 3*).

This tuning heterogeneity was not explained by the physical location of the neurons in the tissue: it persisted within individual shanks of our recording probes (8 shanks spaced 200 µm apart; *Figure 3—figure supplement 4*). Also, the surprising preference for ipsilateral and faint sounds was not explained by the inhibitory contralateral responses: many ipsilateral neurons had evoked responses well above their baseline (insets in *Figure 3D*, see also *Figure 2—figure supplement 2*).

These observations provide an explanation for the population PSTHs in *Figure 2*. In the inactive state, most neurons are similarly tuned, leading to the loud contralateral preference of the average. In the active state, the neural tuning is diverse and, when averaging across neurons, individual differences get averaged out. Across sessions, the fraction of significantly tuned neurons with contralateral preference is positively correlated with CV ($r = 0.70$, $p = 0.0002$, *Figure 3G*), and so is the fraction of neurons with loud preference ($r = 0.71$, $p = 0.0002$, *Figure 3H*).

Inspection of the regression coefficients $\beta_{\mathrm{inter}}$ of the $\mathrm{ILD} \cdot \mathrm{ABL}$ interaction term revealed that the degree of gain modulation across the population is also state-dependent (*Figure 3—figure supplement 5*). One might expect that loud sounds should make ipsilateral cells more ipsilateral, and contralateral cells more contralateral, which would imply a positive correlation between the coefficients $\beta_{\mathrm{ILD}}$ and $\beta_{\mathrm{inter}}$. Scatter plots of these two coefficients across cells show a much stronger correlation in the inactive than in the active state. *Figure 3I* shows the fraction of significantly tuned neurons in the first and third quadrants of this scatter plot as a function of CV. This fraction is positively correlated with CV ($r = 0.66$, $p = 0.0006$, *Figure 3I*). Thus, in the inactive state ABL increases the gain of the tuning of single neurons to ILD, consistent with *Figure 2C*. In the active state, on the other hand, the fraction is approximately 0.5, consistent with an idiosyncratic effect of ABL on ILD tuning. This means that, while ABL might change the tuning to ILD for single neurons, the nature of this change is, across neurons, not consistent with the standard notion of gain modulation.

## Signal and noise correlation matrices

We now turn to the structure and state-dependence of the evoked activity at the population level, which is the main focus of this study.

The regions in firing rate space spanned by the sensory stimuli and those occupied by trial-to-trial variability can be estimated through the eigendecomposition of the signal and noise correlation matrices, respectively (*Figure 4*). In each session, both matrices are of size $p \times p$ where $p$ is the number of recorded neurons. A signal correlation matrix contains correlations between 'tuning curves' of

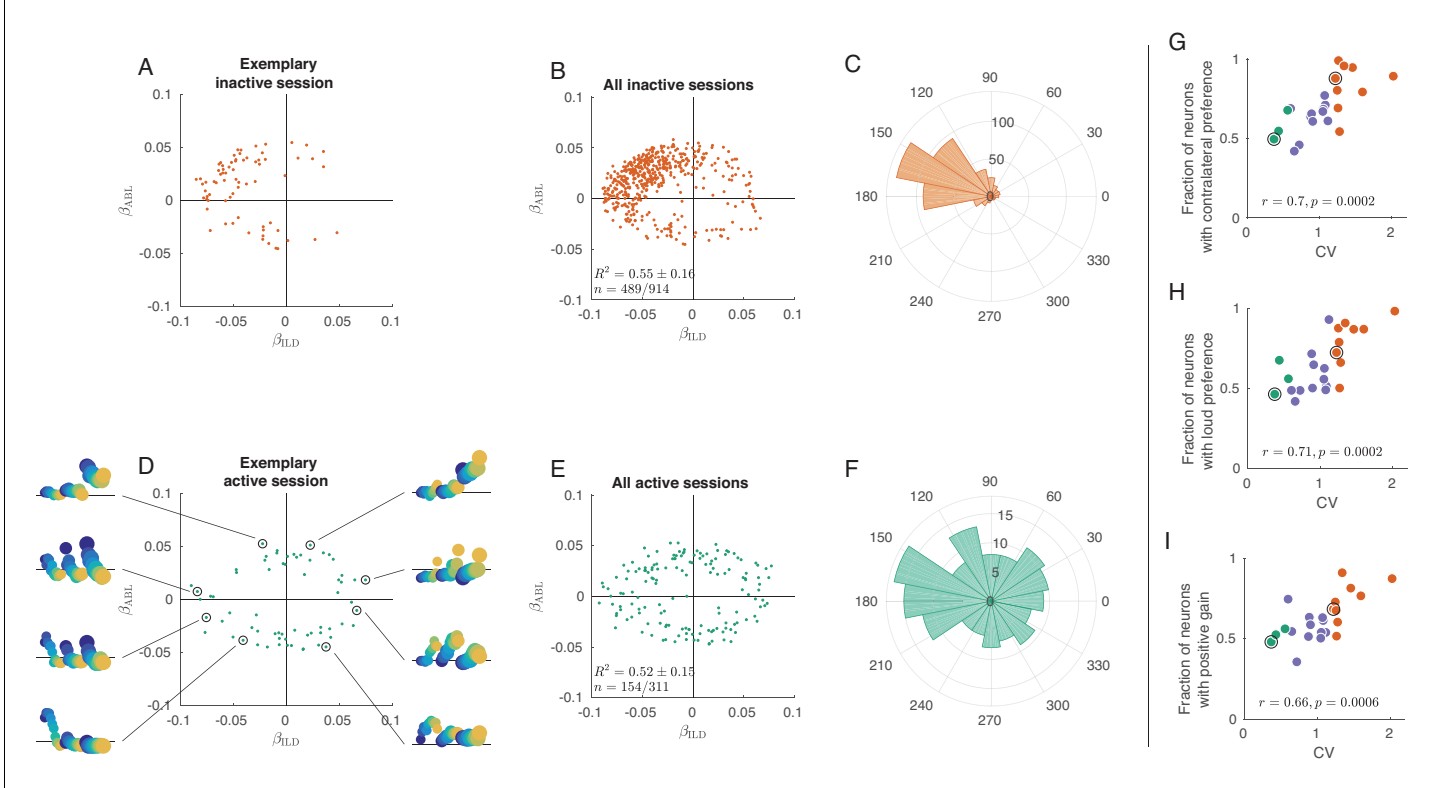

**Figure 3.** Tuning curves in inactive and active states. (A) Single neurons from one exemplary inactive session. The tuning of each neuron was characterized with a linear additive model (see text); horizontal axis shows ILD coefficient and vertical axis shows ABL coefficient. Only neurons with significant tuning ($p<0.01$) are shown. (B) Single neurons ($n = 489$ neurons with significant tuning out of $n = 914$) from nine inactive sessions. $R^2$ values correspond to the average ± standard deviation across the significantly tuned neurons. (C) Circular distribution of the neurons from panel (B). (D) Single neurons from one exemplary active session. Insets show tuning curves of eight exemplary neurons. (E) Single neurons ($n = 154$ neurons with significant tuning out of $n = 311$) from three active sessions. (F) Circular distribution of the neurons from panel (E). (G) Fraction of neurons with contralateral preference (located to the left of the $y$ axis in panels (A) and (D)) across sessions. Green dots are active sessions, orange dots are inactive sessions, black circles mark the two exemplary sessions used in (A) and (D). (H) The same for the fraction of neurons with loud preference (located above the $x$ axis). (I) The same for the fraction of neurons with positive gain modulation (see text).

DOI: https://doi.org/10.7554/eLife.44526.006

The following figure supplements are available for figure 3:

**Figure supplement 1.** Exact analogue of *Figure 3*, but without z-scoring the tuning curves.

DOI: https://doi.org/10.7554/eLife.44526.007

**Figure supplement 2.** Exact analogue of *Figure 3*, but without filtering the neurons based on *p*-value.

DOI: https://doi.org/10.7554/eLife.44526.008

**Figure supplement 3.** Early and late single neuron tuning.

DOI: https://doi.org/10.7554/eLife.44526.009

**Figure supplement 4.** Single neurons from all three active sessions, colored by shanks.

DOI: https://doi.org/10.7554/eLife.44526.010

**Figure supplement 5.** Interaction term in the linear model of single neuron tuning.

DOI: https://doi.org/10.7554/eLife.44526.011

all pairs of neurons (i.e. correlations are computed across trial-averaged responses for $n = 36$ stimuli). An element $(i, j)$ of this matrix is the similarity between the tuning of neurons $i$ and $j$. A noise correlation matrix contains correlations between trial-to-trial fluctuations for all pairs of neurons (here we pool the trials from all stimuli and compute correlations across all $n = 36 \cdot 100$ single trial responses after subtraction of the corresponding mean responses). An element $(i, j)$ of this matrix is the similarity between trial-to-trial fluctuations between neurons $i$ and $j$.

Consider the signal correlation matrices for one exemplary inactive and one exemplary active session shown in *Figure 4A–B*. The neurons are sorted (from up/left to bottom/right) according to the

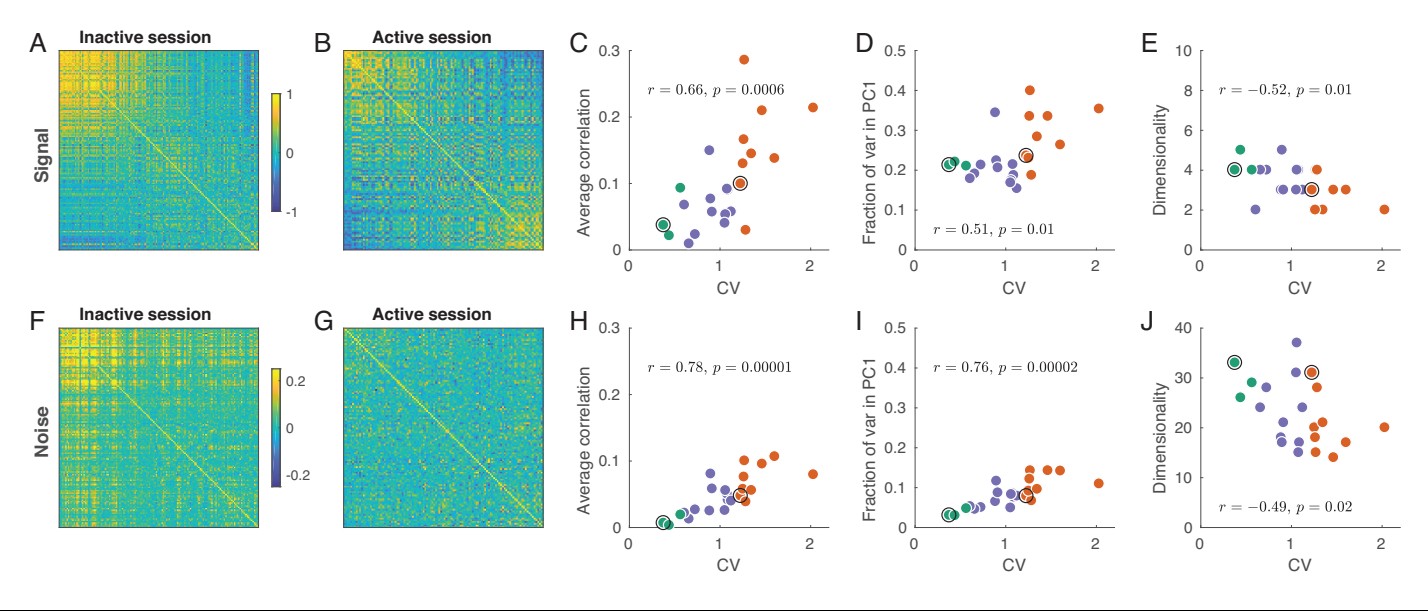

**Figure 4.** Signal and noise correlations in inactive and active states. (**A**) Signal correlation matrix in an exemplary inactive session. Neurons are ordered by their ILD tuning from contralateral (top/left) to ipsilateral (bottom/right). (**B**) Signal correlation matrix in an exemplary active session. Neurons are ordered in the same way. (**C**) Average off-diagonal signal correlation as a function of CV. Here and in the other panels green dots are active sessions, red dots are inactive sessions, black circles mark the two exemplary sessions. (**D**) Fraction of variance explained by the first principal component of the signal correlation matrix as a function of CV. (**E**) Estimated dimensionality of the signal correlation matrix as a function of CV. (**F–J**) The same for noise correlation matrices. Note that the color scale in (**F**), (**G**) is different from (**A**), (**B**).

DOI: https://doi.org/10.7554/eLife.44526.012

value of $\beta_{\mathrm{ILD}}$ from *Figure 3A,D*. In the inactive state, there is a group of similar neurons with contralateral preference, and the remaining neurons do not seem to show any similarity. On the other hand, in the active state there are neurons with similar contralateral preference but also neurons with similar ipsilateral preference. The tuning of these two sets of neurons is anti-correlated.

For the signal correlation matrix in each session, we can compute: (a) the mean correlation, (b) the fraction of variance explained by the first principal component (i.e. $\lambda_{\max}/p$, where $\lambda_{\max}$ is the largest eigenvalue), and (c) the dimensionality. Here, 'dimensionality' refers to the number of significant principal components (out of the total number of 36 PCs). We estimated the dimensionality using Monte Carlo permutations to obtain the distribution of eigenvalues under the assumption of uncorrelated neurons (see Materials and methods). We found that all these three measures were correlated with CV: the more inactive the cortical state, the larger the signal correlations, the larger the first eigenvalue, and the smaller the dimensionality (*Figure 4C–E*). All these effects point to the same underlying phenomenon: as the state progresses from active to inactive, signal correlation matrix 'simplifies' as all neurons become similarly tuned. An alternative analysis using unbiased estimates of the signal correlation matrices yielded the same conclusions (see Materials and methods).

The same analysis for the noise correlation matrices is shown in *Figure 4F–J*. Consistent with previous reports (*Goard and Dan, 2009*; *Renart et al., 2010*; *Ecker et al., 2010*; *Pachitariu et al., 2015*; *Beaman et al., 2017*), we found that the mean pairwise correlation in the active state was around zero (*Figure 4F*) but reached $\sim 0.1$ in the most inactive sessions. The reason for this is well understood (*Renart et al., 2010*; *Ecker et al., 2014*; *Okun et al., 2015*): up-down fluctuations in the inactive state (*Figure 1A*) induce positive correlations between most neurons. Going beyond pairwise analysis, we again see that, with increasing CV, the dimensionality of the noise subspace decreases, while the first eigenvalue of the correlation matrix increases (*Figure 4I–J*). In the active state, the fraction of variance explained by the first PC was not much bigger than $1/p$ (*Figure 4I*), suggesting that the shape of the noise subspace in the active state was approximately (although not perfectly) spherical. Nonetheless, the number of trials in our dataset provided us with enough power to find a large number of significant noise PCs even during the most active states (*Figure 4J*).

## Geometry of the population activity

We saw that both the signal and the noise correlation matrices get similarly re-structured when CV is increasing. To explore how these two processes influence the overall population geometry, we consider $p$-dimensional space where each recorded neuron corresponds to one dimension (*Figure 5A*). The state when all neurons are silent corresponds to the origin of coordinates. Evoked activity in every single trial can be thought of as one point in this space. For each stimulus, there is a 'cloud' of 100 points centered at the mean response. Overall, we have 36 such clouds.

We use principal component analysis (PCA) to define a two-dimensional signal plane that approximately goes through the 36 mean stimulus responses (cloud centers). To do this, we compute an ILD axis as the PC1 direction of the 12 mean stimulus responses after averaging over ABL, and similarly compute an ABL axis as the PC1 direction of the three mean stimulus responses after averaging over ILD. These two axes span a plane, which we define as the signal plane. In our data, this plane is usually very close to the plane spanned by PC1 and PC2 of all 36 mean stimulus responses, but we prefer to use the ILD axis and the ABL axis because it gives an opportunity to explore the relationship between the two.

*Figure 5B* shows the projection of all single trials onto the signal plane in the exemplary inactive session. The loud contralateral stimulus is prominently removed from the rest, corresponding to the single neuron tuning we saw above. At the same time, not much of the ipsilateral tuning can be

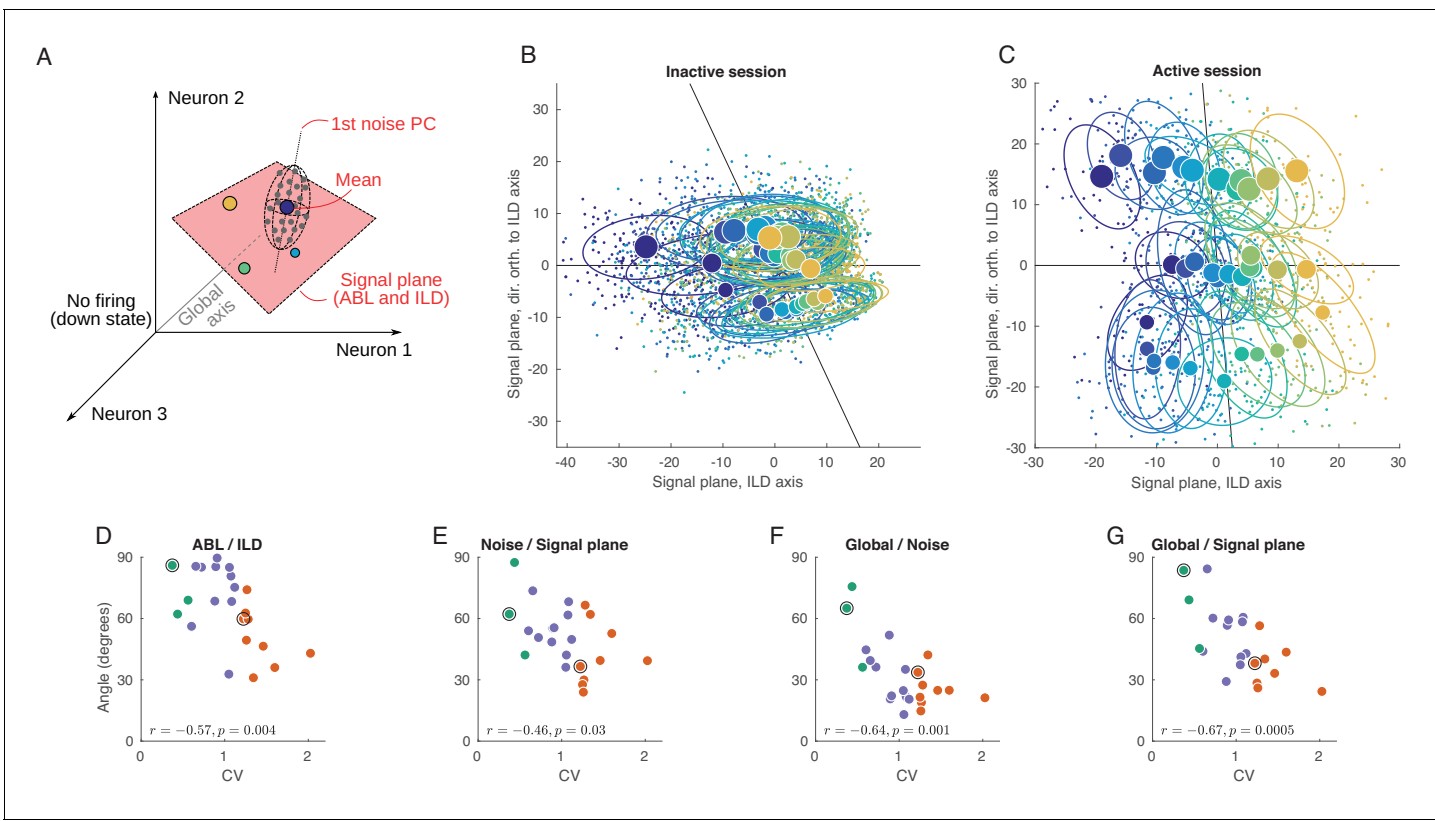

**Figure 5.** Geometry of population activity in inactive and active states. (**A**) Schematic of the population geometry analysis. The firing rate of each neuron forms a dimension, and evoked activity in each trial is represented by a dot in this space. (**B**) Projection of all single trials on the signal plane in one exemplary inactive session. Big dots and ellipses show averages and 50% coverage ellipses for each stimulus. Color coding as in previous figures. Horizontal black line is the ILD axis, the other black line is the ABL axis. (**C**) The same for one exemplary active session. (**D–G**) Relationship between CV and the angles between the ILD axis and the ABL axis (**D**), between the noise axis and the signal plane (**E**), between the noise axis and the global axis (**F**), and between the global axis and the signal plane (**G**).

DOI: https://doi.org/10.7554/eLife.44526.013

The following figure supplement is available for figure 5:

**Figure supplement 1.** The exact analogue of *Figure 5D–G*, but using spike counts in the 0–50 ms window, instead of 0–150 ms.
DOI: https://doi.org/10.7554/eLife.44526.014

seen. The same plot for the example active session is shown in *Figure 5C* and is strikingly different. Here, the stimuli locations closely resemble the grid structure of the used stimuli (cf. *Figure 1C*). The ABL and the ILD axes are nearly orthogonal, unlike in *Figure 5B*.

We systematically assessed how the organization of the code changes as a function of state by computing the angle between several salient axes in firing rate space and by evaluating the dependence of these angles on CV. When interpreting these results, one should keep in mind that, because of the large dimensionality of the space (equal to the number of neurons in a particular recording), random vectors will be close to orthogonal. Angles of approximately 90° should thus be expected if all axes were randomly oriented. Using this approach, we found that, across all sessions, the angle between the ILD and the ABL axes is negatively correlated with CV ($r = -0.57$, $p = 0.004$; *Figure 5D*).

Another important axis in the population representation is the dominant noise axis, which we define as the direction in which the clouds are most stretched (*Figure 5A*). We find it by doing PCA on the pooled noise data (i.e. on all 3600 single-trial points after subtracting the corresponding stimulus means). The orientation of the noise axis relative to the signal plane can show how much the noise interferes with the stimulus representation. Across sessions, the angle between the noise axis and the signal plane is negatively correlated with CV ($r = -0.46$, $p = 0.03$; *Figure 5E*), reflecting a significant overlap between the signal and noise subspaces in the inactive, but not in the active, state. If we analyze the angles between the noise axis and the ILD and ABL axes separately, both are negatively correlated with CV ($r = -0.42$ and $-0.39$). Given that the noise subspace is approximately spherical in the active state (*Figure 4I*), the direction of the principal noise axis is expected to be effectively random in this state, and thus close to orthogonal to the signal plane, as we observe.

We also explored the relationship between the signal and noise subspaces and the 'global' direction, which we define as going from the center of coordinates to the grand mean of all single-trial responses (*Figure 5A*). This is approximately the direction along which the population fluctuates during up-down transitions. The angle between the noise axis and the global axis is also negatively correlated with CV ($r = -0.64$, $p = 0.001$; *Figure 5F*). In the active state, they are close to orthogonal, but in the inactive state up-down fluctuations become the dominant mode of the noise in the evoked responses. The angle between the signal plane and the global axis is negatively correlated with CV as well ($r = -0.67$, $p = 0.0005$; *Figure 5G*), and this remains true for ILD and ABL axes separately ($r = -0.61$ and $-0.61$). This shows that, in the active state, different stimuli evoke approximately the same number of spikes from the population as a whole, whereas in the inactive state different stimuli evoke different numbers of spikes from the population.

We get very similar results if instead of PCA we use demixed PCA (dPCA) that we have previously developed (*Kobak et al., 2016*) to define ABL and ILD axes (with dPCA, the four correlation coefficients in *Figure 5D–G* differed at most by 0.01; see Materials and methods). For the main analysis, we chose PCA for the reason of simplicity. Also, repeating the analysis shown in *Figure 5D–G* using only the early onset responses (0–50 ms instead of 0–150 ms) led to similar conclusions (*Figure 5— figure supplement 1*) with only ABL/ILD angle becoming somewhat less correlated with the CV.

In summary, we saw that the evoked activity in the rat auditory cortex in the inactive state is squeezed along the global axis. However, in the active state all relevant subspaces become close to orthogonal to each other: ABL encoding is nearly orthogonal to the ILD encoding, both of them are nearly orthogonal to the global axis (due to the diversity in tuning across neurons), and the noise axis (as expected from the approximately spherical shape of the noise subspace) is nearly orthogonal to ILD, ABL and global axis. We can see this more directly using a 3D animation (*Video 1*), where the shown three-dimensional subspace is spanned by the signal plane and the global axis. Note that the dominant noise dimension cannot be fully shown because

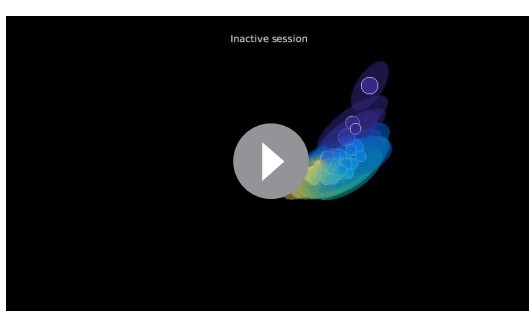

**Video 1.** Three-dimensional subspace spanned by the signal plane and the global axis in the exemplary inactive and in the exemplary active sessions.
DOI: https://doi.org/10.7554/eLife.44526.015

there is only space for three axes in a 3D animation.

## Decoding analysis

How do the state-dependent changes of the population geometry described above influence the amount of information about the stimulus carried by the evoked activity? We can use the decoding of stimulus identity from the population activity to answer this question.

For each session, we trained a binary classifier (decoder) to predict the sign of ILD from neural activity. We chose ILD sign because this allows us to compare the performance of the classifier with the behavioral performance of animals classifying sounds according to ILD (*Pardo-Vazquez et al., 2018*). Furthermore, behavioral studies suggest that ILD discrimination with respect to the midline is more relevant than the estimation of arbitrary ILDs (*Heffner and Heffner, 1988*; *Recanzone et al., 1998*; *Recanzone and Beckerman, 2004*), consistent with the observation that tuning with respect to ILD generally displays a high slope, rather than a peak, at the midline (*Stecker et al., 2005*; *Yao et al., 2013*). We used logistic regression regularized with elastic net and employed nested cross-validation to compute all performance estimates (see Materials and methods). The classifier was trained on trials pooled across all ABLs.

*Figure 6B* shows 'population neurometric curves': classifier performance for each ABL and ILD, averaged over inactive sessions. Individual sessions often yielded noisy curves, but the averages can be very accurately fit with logistic functions. The performance clearly decreases and becomes biased (asymmetric) with decreasing ABL. Because neurons in the inactive state prefer loud contralateral sounds, the overall number of spikes evoked by the population increases with increasing ABL for negative (contralateral) ILDs. The decoder classifies trials with many spikes as contralateral, and so its performance for negative ILDs degrades with decreasing ABL. For positive (ipsilateral) ILDs, there are almost no spikes, which the decoder reliably and correctly classifies as ipsilateral for all ABLs.

Each logistic curve can be conveniently summarized with a 'just noticeable difference' (JND): horizontal distance between points with 75% performance on each side. JND was equal to 9.1 for ABL = 20 dB, to 8.5 for ABL = 40 dB, and to 5.5 for ABL = 60 dB.

*Figure 6B* shows the same analysis for the active sessions. Here, the neurometric curves for three ABLs strongly overlap, are approximately symmetric, and display little bias. JND ranges from 2.5 to 3.4 which is similar to behavioral values measured in other species (*Scott et al., 2007*; *Keating et al., 2013*). We have recently studied ILD discrimination in rats using a subset of stimuli used here with ILDs between ±6 dB (*Pardo-Vazquez et al., 2018*). The average JND of rats was 2.2 dB, thus just slightly smaller than the decoded JND in the active state. Comparisons between behavioral discrimination accuracy and the discrimination accuracy of a decoder from a neural population, however, should be interpreted carefully, as the latter quantity can depend on the number of recorded neurons or time window of analysis. Indeed, we found that the classification accuracy increased with the number of neurons and did not saturate when using all available neurons, either in the active or inactive state (*Figure 6—figure supplement 1*). We also found that the accuracy increased with the length of the spike count window and did not saturate at 150 ms in the active state. In the inactive state, it did tend to saturate at ~50 ms, converging to an ABL-dependent value (*Figure 6—figure supplement 1*).

To quantify classifier performance across ILDs in each session and for each ABL, we computed the average (integral) classification accuracy for all ILDs from −20 to +20 according to the logistic fit (see Materials and methods). *Figure 6C–E* shows these average accuracies for every session and for every ABL. For each ABL, the average accuracy dropped with increasing CV. We summarized the average accuracies using a simple linear model

$$\text{Accuracy} = (a + b \cdot \text{ABL}) + (c + d \cdot \text{ABL}) \cdot \text{CV} + \varepsilon, \tag{2}$$

which was preferred by both AIC and BIC over three independent regressions, one for each ABL (BIC = −327 vs. −320). Linear fits for three ABLs intersected at $\text{CV} \approx 0.3$, indicating that in the active state there was no difference in performance between ABLs. In recordings of the inactive state, on the other hand, the difference between ABLs was pronounced, in agreement with *Figure 6A–B*.

In the above analysis, we used one common classifier for all ABLs in a given session. If instead we train three separate classifiers per session, then the performance increases but only slightly: *Figure 6F* shows the comparison of linear fits for the 'common' and 'ABL-specialized' classifiers. The

difference is absent in the active state, meaning that the sign of ILD can be classified with an ABL-invariant decoder. In the inactive state, the difference was $0.019 \pm 0.023$ (mean $\pm$ SD across nine inactive sessions and 3 ABLs), a small but statistically significant difference ($p = 0.0004$, Wilcoxon sign-rank test).

In the previous sections, we saw that the inactive state is characterized by different neural tuning and by different noise structure. How much of the classification performance decrease can be attributed to each of these two factors? To address this question, we employed a previously suggested shuffling approach (*Averbeck et al., 2006*). We randomly permuted all the trials within each stimulus, separately for each neuron; this procedure effectively sets all noise correlations to zero while preserving the mean responses. After that we ran the same decoding analysis as above (with a single, common decoder for all ABLs), and averaged the results across random shuffles. *Figure 6G* shows the comparison of linear fits for the shuffled and non-shuffled cases. The difference was absent in the active state, while performance increased after shuffling in the inactive state ($0.025 \pm 0.028$, $p = 0.0004$). In other words, the noise correlations in the inactive state impair the accuracy of the code, consistent with our observations above that the noise subspace is aligned with the stimulus subspace in the inactive state.

This finding does not necessarily imply that the presence of noise correlations affects the optimal decoding strategy, as has been pointed out before (*Averbeck et al., 2006*). To check this, we performed trial shuffling only on the training sets, while preserving the testing sets intact (*Averbeck et al., 2006*). *Figure 6H* shows the comparison of linear fits for the shuffled and non-shuffled cases. Once more, the difference was absent in the active state, but performance noticeably decreased in the inactive state ($-0.050 \pm 0.047$, $p = 0.0003$). These analyses suggest that, while the signal and noise subspaces display a significant degree of collinearity in the inactive state, they are still not fully aligned.

Decoding performance in the inactive state after shuffling remains clearly worse than performance in the active state (*Figure 6G*), suggesting that tuning curve reorganization also contributes strongly to the state-dependence of ILD discrimination. The fact that many neurons in the active state, but not in the inactive state, are tuned to ipsilateral ILDs, raises the question of whether this increased discrimination ability is simply due to an ability to represent ipsilateral sounds in the active state. To address this question, we assessed the accuracy of ILD estimation (as opposed to discrimination with respect to the midline) separately for ipsi- and contralateral sounds across states. This analysis revealed that estimation accuracy still degrades with increasing CV for both ipsi- as well as contralateral ILDs, even when they are considered separately (*Figure 6—figure supplement 2*).

## Discussion

We have described the state-dependence of population activity in the rat auditory cortex at three different levels: the average population response, the responses of single neurons, and the geometrical structure of the high-dimensional population responses. Our stimulus set was two-dimensional, with one dimension, ABL, varying along a prothetic continuum (different stimulus intensity), and the other, ILD, varying along a metathetic one (different lateralization at constant intensity). One can expect that the intensity of the neural population response reflects the intensity of the stimulus; at the same time, the intensity of population activity is strongly affected by the brain state, with inactive states causing global up-down fluctuations of this intensity. Thus, our stimulus set is particularly useful to explore the state-dependence of the representation of intensity, and of the effect of intensity on other tuning dimensions (typically described as changes in gain). Both these issues have so far remained largely unexplored. That said, our stimulus set is still low-dimensional, preventing us from studying the state-dependence of the geometrical structure of high-dimensional stimulus spaces such as natural images or sounds (*Stringer et al., 2018*). Nevertheless, *Pachitariu et al. (2015)* also found more accurate representations of natural sounds in the desynchronized state.

### State-dependence of neural tuning to the stimulus

The canonical picture of neural tuning in auditory cortex is that neurons within a given hemisphere tend to prefer contralateral sounds (*Stecker et al., 2005*; *Campbell et al., 2006*; *Keating et al., 2013*; *Lui et al., 2015*; *Keating et al., 2015*). In rat, in particular, it was recently reported that all recorded neurons in one study had a contralateral preference (*Yao et al., 2013*). In terms of sound

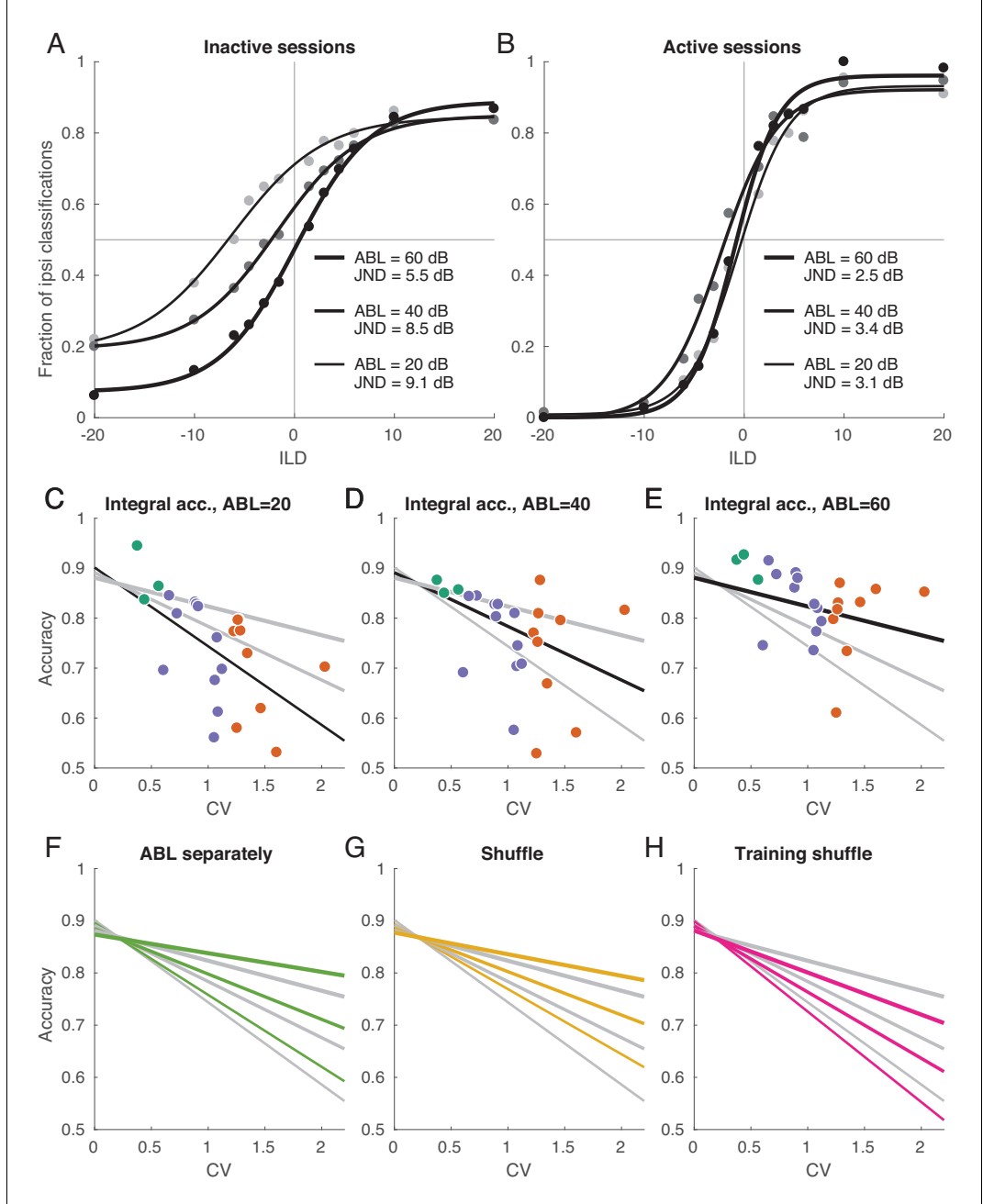

**Figure 6.** Decoding analysis in inactive and active sessions. (A) Population neurometric curves in inactive sessions. Dots show the fraction of ipsilateral classifications when classifying the sign of ILD (from grey to black, ABL = 20 dB to ABL = 60 dB), averaged across inactive sections. Curves show logistic fits. (B) Population neurometric curves in active sessions. (C–E) Integral classification accuracy for $\text{ILD} \in [-20, +20]$ across sessions for each ABL separately. Gray lines show constrained linear fits for each ABL. Black lines highlight the fit corresponding to the respective scatter plot. (F) The change of linear fits when one uses separate decoder for each ABL (green). (G) The change of linear fits after shuffling trials for each neuron in each condition (yellow). (H) The change of linear fits after shuffling the training sets only (magenta).

DOI: https://doi.org/10.7554/eLife.44526.016

The following figure supplements are available for figure 6:

**Figure supplement 1.** Classification accuracy as a function of spike count window length and of the number of neurons.
DOI: https://doi.org/10.7554/eLife.44526.017

**Figure supplement 2.** We used linear decoding of ILD, separately for the contralateral and ipsilateral sounds and separately for each ABL.
DOI: https://doi.org/10.7554/eLife.44526.018

intensity, the canonical V-shaped frequency-amplitude tuning curve present already at the level of the auditory nerve and in subsequent processing stages (*Moore et al., 2010*; *Schnupp et al., 2011*) implies a preference for loud sounds, although this picture describes only the typical trend, and neurons with non-monotonic tuning to intensity can be found throughout the auditory pathway, including the periphery (*Davis et al., 1996*; *Greenwood and Maruyama, 1965*; *Aitkin, 1991*; *Wu et al., 2006*).

Since our recordings came from a single hemisphere, this literature strongly suggests that we should observe a clear preference of the population-averaged activity for large ABLs and negative ILDs. We find that this expectation holds true in the inactive state (*Figure 2A–C*) but that, surprisingly, it breaks down as the cortex becomes strongly activated (*Figure 2D–G*). Examination of single-cell tuning curves revealed that this phenomenon is not due to the loss of tuning under cortical activation. Rather, it is due to the appearance of tuning diversity. Our analysis revealed an approximately symmetric preference for ipsi-, contra-, quiet and loud stimuli across the neural population in the most active states (*Figure 3D–H*). The state-dependent tuning to stimulus lateralization has the significant implication that the oft-observed preference for contralateral sounds in the auditory cortex is not a hard-wired property.

A possible explanation for the previously unreported flat tuning of the population-averaged firing rate to ILD and ABL is that the activation/inactivation continuum might not have been sampled with sufficient depth in the past. For example, in *Yao et al. (2013)*, no explicit quantification of brain state was provided, although the authors state that recordings took place under 'deep' anesthesia. Most studies of ILD coding were conducted under anesthesia, and the anesthetized brain is typically inactivated. In contrast, results more similar to what we observed in the active state have been reported in unanesthetized animals, for example with respect to non-monotonic level tuning (*Evans and Whitfield, 1964*; *Young and Brownell, 1976*; *Pfingst and O'Connor, 1981*; *Davis et al., 1996*; *Sadagopan and Wang, 2008*). Urethane anesthesia — used in our study — is well known to be compatible with a variety of levels of activation (*Murakami et al., 2005*; *Clement et al., 2008*; *Renart et al., 2010*), although, depending on the protocol, this variety is not always observed. Our results emphasize the importance of taking the brain state prevalent during an experiment into account when studying the neural code (*Harris and Thiele, 2011*; *Zagha and McCormick, 2014*).

Beyond single-cell analysis, we have characterized signal and noise correlation matrices from neural population data, which previous theoretical work has shown to be critical for understanding the accuracy of population codes (*Panzeri et al., 1999*; *Sompolinsky et al., 2001*; *Averbeck et al., 2006*; *Moreno-Bote et al., 2014*). The structure of the signal subspace confirms intuitions derived from neural tuning curves. The overall preference of most neurons for loud, contralateral sounds in the inactive state is reflected in the overlap between the ILD, ABL and global activity axes during cortical inactivation. The signal subspace in the most active states is nearly orthogonal to the global activity axis, consistent with the flat tuning of the population firing rate (*Figure 2*). More surprisingly, the approximately symmetric and idiosyncratic tuning to ABL and ILD in this state furnaces a nearly orthogonal representation of the two dimensions at the level of the population (*Figure 5C*). Although ILD and ABL are 'orthogonal' stimulus dimensions (*Figure 1C*), this finding is far from trivial because all our neurons are recorded from the same hemisphere. As we argued above, one could have expected that populations containing neurons from both hemispheres would be needed to reveal explicitly the differential nature of the ILD code (as described previously in *Keating et al., 2015*). Since neurons in the auditory pathway before the superior olive are uni-lateral, this kind of representation — faithful to the structure of the stimuli themselves (compare *Figures 1C* and *5C*) — is constituted de novo and becomes available during cortical activation. The possibility of observing the full differential ILD code in a single hemisphere is consistent with lesion studies showing that ILD estimation close to the midline is still possible under uni-lateral inactivations of the auditory cortex (*Malhotra and Lomber, 2007*; *Malhotra et al., 2008*).

## State-dependence of ILD-tuning at different sound levels

We explored the way in which stimulus intensity affects the coding of stimulus identity by quantifying how the interaction between ILD and ABL is represented by neural populations in the auditory cortex. We found that the effect of ABL on ILD tuning is also significantly shaped by the brain state. Stimulus intensity enhanced contralateral tuning during the inactive state both at the level of the

population-averaged activity (*Figure 2C*) and at the single cell level (*Figure 3—figure supplement 5D–E*). Similar changes in gain have been observed in tuning curves to sound location (see e.g. Figure 6 in *Stecker et al., 2005*) driven by changes in sound intensity and in tuning curves to the orientation of visual gratings (*Skottun et al., 1987*) driven by changes in stimulus contrast. If the same pattern held during cortical activation, one would see a strong correlation between the sign of the ILD term and the sign of the interaction term in *Equation (1)*. These two quantities were, however, only weakly related across the population in the active state (*Figure 3I* and *Figure 3—figure supplement 5F*). At the population level, gain modulation in the inactive state is reflected in the overlap between the ILD and the ABL axes (*Figure 5B,D*), and in the level-dependence of the accuracy and bias of the decoders of sound lateralization (*Figure 6B, C–F*). At the same time, the lack of a consistent pattern of gain modulation across the population in the active state results in the faithful orthogonal stimulus representation shown in *Figure 5C,D* and in the level-invariant accuracy of our lateralization decoders (*Figure 6B, C–F*), similar to what was observed behaviorally (*Recanzone and Beckerman, 2004*; *Stellmack et al., 2004*; *Nodal et al., 2008*; *Pardo-Vazquez et al., 2018*). Presumably, if tuning curves for ILD were gain modulated in the standard fashion during the active state, the ILD and ABL axes in *Figure 5C* would still be approximately orthogonal, but the separation between the centroids for each ABL (the three approximately horizontal rows of large points in this figure) would grow with ABL. This would possibly lead to an increase in decoding accuracy with increasing ABL, unlike what is observed behaviorally.

The fact that, behaviorally, ILD discrimination is level-invariant has sparked interest in finding neurons representing ILD in a way that is *explicitly invariant* to sound level. Such explicitly invariant representations are generally not found (*Park et al., 2004*; *Tsai et al., 2010*; *Kyweriga et al., 2014*). Opponent-process models (*Stecker et al., 2005*; *Tsai et al., 2010*; *Keating et al., 2015*) suggest that explicitly invariant representations are not necessary for a functionally invariant representation. In these models, a feature can be invariantly represented by a population if the effect of intensity is the same on neurons oppositely tuned to the feature, so that the effect of intensity will not be present on the difference between the activity of these oppositely tuned cells. The mechanism for level invariance in our data during cortical activation is similar in spirit, although less organized, as the effect of ABL on ILD tuning is generally unstructured (*Figure 3I* and *Figure 3—figure supplement 5F*). Invariance to certain stimulus features by means of approximately orthogonal representations between these and other 'nuisance' features across a neural population, has also been highlighted previously in the context of invariant object recognition in the ventral visual pathway (*DiCarlo and Cox, 2007*; *Pagan et al., 2013*; *Rust, 2014*).

## State-dependence of trial-to-trial variability

Changes in brain state have been shown to have a large effect on trial-to-trial variability and temporal fluctuations in neural activity (*Poulet and Petersen, 2008*; *Renart et al., 2010*; *Ecker et al., 2014*). Several studies addressed this issue by quantifying the effect of brain state on pairwise correlations in spike counts. These studies found that the mean pairwise correlation decreases with the level of cortical activation, sometimes to negligible levels. In turn, the positive correlations found during cortical inactivation reflect global coherent activity fluctuations across the population (*Goard and Dan, 2009*; *Renart et al., 2010*; *Ecker et al., 2010*; *Ecker et al., 2014*; *Okun et al., 2015*; *Pachitariu et al., 2015*; *Beaman et al., 2017*). Our results confirm these previous findings (*Figure 4H*).

At the population level, we found that the noise subspace becomes more anisotropic as the cortex becomes more inactivated. Moreover, the overall orientation of the noise subspace tends to align with the global activity axis, along which all neurons are modulated coherently. This noise structure impairs the representation of stimulus dimensions that also carry the population activity along the global axis. Because in the inactive state this is true for both ILD and ABL, the neural representation of ILD suffers and becomes ABL-dependent, as confirmed by our decoding results (*Figure 6*).

The near-zero mean correlation found in recordings of the most active states (*Figure 4H*) is consistent with both a spherical noise subspace (i.e. the dashed ellipsoid in *Figure 5A* looking more like a sphere), or with the existence of preferred noise directions that are approximately orthogonal to the diagonal direction (*Montijn et al., 2016*). Our data suggest the former scenario. This means that spiking correlations do not significantly constrain the accuracy of the population code during the

active state. Consistent with this finding, we found effectively no difference between the accuracy of sound lateralization decoders using the actual data, or trial-shuffled data (*Figure 6G*), in this state.

## Time-course of stimulus representations during the evoked response

In this study, we focused on quantifying the state-dependence of neural codes using spike counts, thus leaving for future analyses the important problem of the detailed temporal structure of the code. Nevertheless, inspection of the results using spike count windows of different lengths or centered at different time points already revealed some interesting findings.

Brain state had a large impact on the structure of late sensory responses. Only in the active state the number of significantly tuned neurons increased from the onset (0–50 ms) to the late (100–150 ms) period, which likely explains why decoding accuracy grew monotonically during the evoked response in this state, whereas it saturated after the onset in recordings during inactive states (*Figure 6—figure supplement 1A–C*). Many neurons in the active state displayed inhibitory responses (below baseline, see *Figure 2—figure supplement 2*), especially in the late period. The approximate balance of excitatory and inhibitory responses was largely responsible for the weak modulation of the population-averaged response in the active state (*Figure 2F*) which dropped slightly below baseline in the late period (*Figure 2E*).

These findings raise the issue of whether the strong reorganization of the code across states that we have described is present already in the onset response, which typically receives more attention in auditory physiology. Our analyses suggested that this is indeed the case. At the single neuron level, the heterogeneity (and lack thereof) in tuning in the active (inactive) state is maintained in the onset period (*Figure 3—figure supplement 3*). At the population level, the state dependence of the geometry of the population code is also maintained qualitatively if only onset responses are considered (*Figure 5—figure supplement 1*). Thus, while late responses clearly contribute to the state-dependence of the full evoked response, onset responses contribute to it in a similar fashion.

## Mechanisms of state-dependence

What mechanisms could underlie the state-driven reorganization of sound-evoked responses that we have observed? Many studies have suggested that changes in the prevalence of slow global spiking coordination within cortical circuits can be mediated by changes in the effective level of 'adaptation' present in the circuit (*Latham et al., 2000*; *Bazhenov et al., 2002*; *Compte et al., 2003*; *Parga and Abbott, 2007*; *Curto et al., 2009*; *Sanchez-Vives et al., 2010*; *Mochol et al., 2015*). The key dynamical property of 'adaptation' is to generate slow negative feedback in the circuit dynamics. Thus, many different biophysical mechanisms could be responsible for it, including for instance slow activity-dependent hyperpolarizing currents (*Compte et al., 2003*), synaptic depression (*Bazhenov et al., 2002*), or feedback inhibition mediated by GABA-B receptors (*Sanchez-Vives et al., 2010*).

Although the activation/inactivation continuum as we defined it here is one-dimensional, the space of global network regimes which we summarize as 'brain state' is surely higher dimensional. Even at the level of neuromodulation, cholinergic, noradrenergic and serotoninergic influences can have independent effects on cortical dynamics (*Vanderwolf, 1988*; *Constantinople and Bruno, 2011*; *Polack et al., 2013*), and different cell types can be responsible for (presumably different forms of) cortical desynchronization (*Fu et al., 2014*; *Chen et al., 2015*). Nevertheless, *Clement et al. (2008)* found that state transitions under urethane resemble those observed between slow-wave and REM sleep, and have the same cholinergic origin. Activation of the nucleus basalis, directly or indirectly, desynchronizes the cortex (*Goard and Dan, 2009*; *Sakata and Harris, 2012*; *Harris and Thiele, 2011*). A recent study found that ACh-induced cortical activation is locally mediated by somatostatin (SOM) interneurons in visual cortex (*Chen et al., 2015*). Changes in the interplay between recurrent and feedforwad excitation and local inhibition could mediate the diversification of tuning profiles we have observed, specially taking into account that many stimulus-specific responses are inhibitory in the active state (*Figure 2—figure supplement 2*). This is also consistent with recent theoretical studies suggesting that changes in local inhibition are the critical factor underlying differences in brain dynamics across different states of cortical activation (*Stringer et al., 2016*) or during attention (*Kanashiro et al., 2017*). The established role of ACh in mediating the effects of attention (*Harris and Thiele, 2011*; *Thiele and Bellgrove, 2018*) suggests a

possible link between attention-driven desynchronization (*Cohen and Maunsell, 2009*) and the desynchronized activity we have studied.

## Functional implications

What are the functional consequences of the different coding schemes that we observe across brain states? The emphasis on difference (as opposed to sum) modes that is apparent in the most active states seems to be well suited for sensory representations supporting a process of discrimination. Decision variables in sensory discrimination tasks are posited to involve differences in activity between oppositely tuned populations of neurons (*Gold and Shadlen, 2002*; *Machens et al., 2005*; *Brunton et al., 2013*; *Pardo-Vazquez et al., 2018*). Indeed, our decoders for ILD end up subtracting the activity of neurons with coefficients $\beta_{\mathrm{ILD}}$ of different signs (fraction of non-zero decoder weights that have the same sign as $\beta_{\mathrm{ILD}}$ was $0.81 \pm 0.08$, mean $\pm$ SD across sessions). As in engineering, this implements a process of common-mode-rejection that will render the discrimination process robust to overall changes in excitability, or any sources of noise causing common fluctuations in both relevant populations. Consistent with this interpretation, brain activation is seen during behavioral states of attentive wakefulness and active information sampling (*Vanderwolf, 2003*; *Castro-Alamancos, 2004*; *Gervasoni et al., 2004*; *Poulet and Petersen, 2008*; *Harris and Thiele, 2011*; *Zagha and McCormick, 2014*) during which the brain is presumably actively engaged in identifying the fine structure of the environment.

The functional role of the population structure in the inactive state is more uncertain. Brain inactivation is observed during immobility or 'automatic' behaviors during wakefulness (*Vanderwolf, 2003*; *Castro-Alamancos, 2004*; *Gervasoni et al., 2004*; *Poulet and Petersen, 2008*; *Harris and Thiele, 2011*; *Zagha and McCormick, 2014*), and during slow wave sleep (*Steriade et al., 1990*; *Steriade et al., 1993*; *Steriade and McCarley, 2013*). These observations, together with the fact that the strong global temporal fluctuations observed during this state have an internal origin (*Sanchez-Vives and McCormick, 2000*; *Poulet et al., 2012*), suggest that perhaps inactivated dynamics support an internal role, that is a role not directly related to the interrogation of the environment. Data showing that cortical upstate-onsets coincide with hippocampal sharp-wave-ripples (*Battaglia et al., 2004*), which are important for memory consolidation, support this view.

Recent studies quantified the effect of rapid, trial-by-trial changes in the degree of desynchronization in auditory cortex during auditory detection in mice (*McGinley et al., 2015a*; *McGinley et al., 2015b*). They found that states of maximal synchronization and desynchronization, both led to errors (misses and false alarms respectively), with an intermediate state being associated with the best performance. These results suggest that the overall 'excitability' of the tissue might increase with the level of desynchronization. It will be of interest to explore how these findings generalize to sensory discrimination tasks where, by definition, the tendency to response is decoupled from correct performance.

## Conclusions

We showed that the cortical activation continuum is associated with a complete reorganization of the geometry of the population code that effectively eliminates all influence of the average firing rate during states of strong cortical activation. This means that, for these states, all sources of variability (ILD, ABL and noise) are reflected in differences in the identity of the responding neurons, and not between the overall number of spikes fired by the population as a whole. As the cortex becomes more inactive, this identity code becomes progressively replaced by an intensity code, in which the total number of spikes fired by the population reflects both the ILD and ABL of the stimulus and trial-to-trial variability. The massive reorganization of sensory activity across levels of cortical activation that we have demonstrated in our study highlights the importance of a more detailed understanding of the cognitive/behavioral processes associated with the activation continuum and the functional role of brain state in general.

## Materials and methods

### Electrophysiology

Surgery and anesthesia

All experimental procedures in this study, covered under project #2018/007 'Principles of Sensory- and Memory-guided Auditory Decision Making: Behavior and Neural Basis', have been approved by the Champalimaud Foundation's animal welfare body.

Eighteen adult female Long-Evans rats (240–400 g; Charles River, Italy) were used for electrophysiological recordings in this study. Seven animals were anesthetized with 1.65 g/kg urethane (solution at 25%, ip) and the rest with 1.35 g/kg urethane (solution at 20%, ip). All were given Atropine Methil-Nitrate (0.5 mg/Kg ip); Dexamethasone (1.2 mg/Kg ip); and saline (1 ml ip). Immediately before surgery Ketamine-Xylazine (1 ml/Kg of a solution containing 75 mg/ml of ketamine and 5 mg/ml of xylazine, ip) was administered, the rat was placed in a stereotaxic frame and 0.2 ml of lidocaine was injected sc in the scalp. Core body temperature was monitored with a rectal thermometer and maintained at 37°C with an electric heating pad. The temporal muscle was partially removed and a screw was implanted in the anterior part of the skull, using bone screws and acrylic cement. Then the head of the animal was held using the implanted screw and the mouthpiece and the ear bars of the stereotaxic were removed. Following stereotaxic coordinates and external landmarks in the temporal bone (Ogawa et al., 2011), a 3 × 2 mm window was open in the skull on top of the auditory cortex. The dura was removed under a dissection microscope and the probe was positioned, with a manual micromanipulator, perpendicular to the brain surface, avoiding blood vessels. The skull cavity was kept moist with saline during this process.

Experimental setup and stimulus generation

System three equipment from Tucker-Davis Technologies (TDT, Alachua, FL), controlled by a personal computer running OpenProject (TDT), was used for stimulus presentation and data acquisition. The recordings were made in a double-wall anechoic chamber. The head of the rat was supported from the front and the area around the ears was unobstructed. Binaural sounds were presented from two loudspeakers (Knowles model number 2403 260 00029) placed in front of the ear canals at 0.5 cm distance. The speakers were calibrated before each experiment to flatten (±2 dB SPL) their frequency response, using a Brüel and Kjaer Free-field $\frac{1}{4}$-inch microphone. Stimuli were 150 ms broadband (5–20 kHz) noise burst with 5 ms raised-cosine onset/offset ramps generated by a TDT RZ6 module, with 24 bit precision at a 50 kHz sampling rate.

Experimental procedure

Extracellular recordings were performed with 64 channel (8 shanks × 8 sites) silicon probes (Buzsaki64, NeuroNexus, Ann Arbor, MI). The probe was connected to an analog headstage, which sent the data to a PZ5 module (TDT). Raw electrophysiological data at a 24 kHz sampling rate was recorded with a RZ2 module (TDT) and stored, together with stimuli information and time stamps, in a computer for off-line analysis. The probe was advanced manually while visualizing the neural activity, and the final depth was adjusted to maximize the number of channels showing spikes and clear auditory evoked responses in the local field potential. Then, the craneotomy was filled with warm agarose at 5% in saline solution to protect the exposed cortex and provide lateral support to the probe shanks. The recording protocol included 30 min of spontaneous activity followed by the stimulation period, which lasted for about 1 hr. Noise bursts were presented at different intensities to each ear (interaural level difference, ILD) to create a percept of lateralization and the effect of sound intensity was tested by using different average binaural levels (ABLs). A total of 36 combinations (12 ILDs × 3 ABLs) were presented. ILDs were ±1.5, ±3, ±4.5, ±6, ±12, and ±20 dB; ABLs were 20, 40 and 60 dB SPL. Negative ILD values correspond to stimuli with higher intensity to the left ear (contralateral to the recording hemisphere). Each combination was presented 100 times, in pseudorandom order, with all 36 combinations being presented every 36 trials (except for rat #1 where stimuli were chosen randomly; this rat had 67–138 repetitions per stimulus, with the same 3600 overall). The inter-stimulus interval changed randomly between 850 and 1150 ms across presentations.

## Spike sorting

Single units were isolated by a semi-automatic algorithm (https://sourceforge.net/projects/klustak-wik) followed by manual clustering (https://github.com/klusta-team/klustaviewa), as described in *Rossant et al. (2016)*. Single units selected for further analysis had less than 10% contamination in an absolute refractory period of 2 ms.

## Quality control and data preprocessing

We obtained 22 recording sessions from 18 animals. The recordings in one animal (#6) showed so little evoked activity that we excluded its sessions from further analysis. Among the remaining 20 sessions, several showed marked changes in the amount of cortical activation during the recording. We manually split three sessions into more active and more inactive parts, obtaining $n = 23$ 'sessions' that were analyzed downstream. These sessions had $104 \pm 23$ (mean $\pm$ SD) isolated neurons (min 67, max 174).

## Data analysis

The full MATLAB code for the analysis is located at https://github.com/dkobak/a1geometry (*Kobak, 2019*; copy archived at https://github.com/elifesciences-publications/a1geometry). We made the spike count data (spike counts for each neuron for each stimulus presentation from $-50$ ms to $150$ ms in 50 ms bins) available in the same repository. This allows to reproduce most of our figures. The complete dataset that was collected, including spike time data not analyzed here, is available upon reasonable request to the corresponding author.

## Statistical testing

All reported p-values for Pearson's correlation coefficients are computed using $t$-distribution. For paired tests we used Wilcoxon sign-rank test. We did not use power analysis to plan the sample size because our study is exploratory, not confirmatory. All reported p-values should be interpreted in this context.

## Measures of activation

We used a sliding (without overlaps) window of 30 s to compute measures of cortical activation. Inside each window, we used a sliding (without overlaps) sub-window of 20 ms and counted the total number of spikes (from all neurons) in each sub-window. If this number was 0, this sub-window was counted as a down state. We computed a fraction of down states as the fraction of down-state sub-windows in each window, and coefficient of variation (CV) as the standard deviation of spike counts divided by the mean spike count across all sub-windows in each window. We inspected the fraction of down states and the CV across time in each session, and manually split three sessions where they changed a lot (see above). *Figure 1D* shows the average fraction of down states and the average CV across all windows in each session. We use the average CV as our measure of activation for all downstream analysis.

For the conditioning analysis (*Figure 2—figure supplement 1*), we split all stimuli presentations into those with no spikes from any of the neurons in the 50 ms window before stimulus onset (conditioning on down state) and the rest (conditioning on the up state). The rest of the analysis is fully explained in the Results section.

## Analysis of correlation matrices

Let the number of neurons recorded in a session be $p$. The number of different stimuli is 36 and the total number of stimuli presented during each session is $n = 3600$ (except for the sessions that were split in two parts, as described above). The starting point of this analysis is $n \times p$ matrix of evoked responses averaged across the 150 ms window after stimulus onset.

To obtain the stimulus correlation matrix, we averaged the responses of each neuron to each stimulus over the repeated presentations of this stimulus, collected them into a $36 \times p$ matrix $X$, standardized each column and computed the correlation matrix $X^{\top}X/36$. This procedure can potentially introduce a bias towards the noise correlations. We have additionally used an unbiased estimate of signal correlations, splitting all trials into two parts, averaging over them separately to obtain $X_1$ and

$X_2$, and estimating the signal correlation matrix as $X_1^\top X_2 / 36$ (*Stringer et al., 2018*). This did not affect our conclusions.

To obtain the noise correlation matrix, for each neuron and each stimulus, we subtracted the average response over repetitions from each of the repetitions. We collected the resulting trial-to-trial deviations into a $n \times p$ matrix $X$, standardized each column and computed the correlation matrix $X^\top X / n$. We used a sliding window of 11 trials (for each stimulus type) to subtract a running average, so that the noise correlation matrix were not influenced by any slow changes in tuning that could happen within a session. This had only a minor effect on *Figure 4* (compared to using the global average).

We estimated the dimensionality of these matrices (*Figure 4E and J*) using a permutation-based Monte Carlo approach. On each of the 100 Monte Carlo iterations, we permuted the values in each column of the $X$ matrix, independently in each column. This sets all population correlations to zero. We then computed the correlation matrix and its eigenvalues, sorted the eigenvalues in the decreasing order, and found 99th percentile across Monte Carlo repetitions. The dimensionality was defined as the number of the original eigenvalues (when sorted in the decreasing order) that were larger than the corresponding Monte Carlo percentile value.

## Stimulus and noise subspaces

We again average the responses of each neuron to each stimulus over the repeated presentations of this stimulus, collecting the resulting mean responses into a $36 \times p$ matrix $X$. We define ILD axis as the first principal component direction of $X$ after it is averaged over ABL (yielding a $12 \times p$ matrix). Similarly, we define ABL axis as the first principal component direction of $X$ after it is averaged over ILD (yielding a $3 \times p$ matrix). We define signal plane as the plane spanned by these two axes. An angle between a plane A and a vector B was defined as the angle between B and its projection on A.

When projecting the single-trial data onto the signal plane in *Figure 5B–C*, we subtracted the mean response over all stimuli (mean of the $36 \times p$ matrix $X$) from each single trial.

For demixed PCA (*Kobak et al., 2016*), we represented the $n \times p$ matrix as a four-dimensional array $X_{ijkl}$ with a separate dimension for neuron id, ILD, ABL, and stimulus repetition number. Briefly, dPCA decomposes population activity into ILD components, ABL components, ILD·ABL interaction components, and the noise, and finds a decoder and an encoder for each component. A decoder is a linear readout that allows to obtain a component from the population activity and an encoder is an axis that shows how strongly each neuron is expressing this component. We used MATLAB dPCA implementation from https://github.com/machenslab/dpca and used built-in cross-validation to get the optimal regularization parameter. The encoding axis of the leading ILD component was always very similar to the ILD axis defined above, and the same was true for the ABL axes. We recomputed the correlations shown in *Figure 5D–G* using these two dPCA axes and obtained practically identical results (all differences in correlation coefficients within 0.01).

## Decoding

All decoding (classification/regression) analyses were run for each session separately. We used either logistic regression (for binary classification) or linear regression with elastic net regularization from the glmnet MATLAB library (*Friedman et al., 2010*). The $\alpha$ parameter that determines the balance between the ridge and the lasso regularization penalties was fixed at $\alpha = .5$. We used nested cross-validation to choose the optimal value of the regularization parameter $\lambda$ and to obtain an unbiased estimate of the performance. The outer loop was 10-fold cross-validation. The inner loop was performed using the cvglmnet function that does 10-fold cross-validation internally. We then used the value of $\lambda$ that yielded minimum cross-validation loss (lambda_min) to make the test-set predictions. Outer-loop performance was assessed via test-set accuracy for binary classification and test-set $R^2$ for regression. We trained the following decoders:

1. Binary classification of the ILD sign (*Figure 6A–E*).
2. Binary classification of the ILD sign, trained for each ABL separately; three classifiers (*Figure 6F*).
3. Binary classification of the ILD sign after shuffling the trials within each of the 36 stimuli for each of the neurons separately. We used 10 shuffles (*Figure 6G*).

4. Binary classification of the ILD sign after shuffling the trials as above but only on each training set, leaving each test set intact. We used 10 shuffles (*Figure 6H*).
5. Time-resolved binary classification of the ILD sign using neural activity averaged over $[0, T]$ window after stimulus onset, for $T$ ranging from 10 ms to 150 ms in steps of 10 ms; 15 classifiers (*Figure 6—figure supplement 1*).
6. Binary classification of the ILD sign using randomly subsampled set of neurons. We used the number of neurons from 10 to $N$ in steps of 10, where $N$ is the number of neurons in a session. This yields $\lceil N/10 \rceil$ steps. We used 10 random samples for each step, making up $10 \cdot \lceil N/10 \rceil$ classifiers in total (*Figure 6—figure supplement 1*).
7. Linear regression of ILD value, separately for the ipsilateral ILDs and for contralateral ILDs; two regression models (*Figure 6—figure supplement 2*).

## Assessing performance of binary classifiers

For every binary classifier that we trained, we assessed its performance for each ILD/ABL combination (meaning that we computed the test-set accuracy for each ILD/ABL stimulus separately). For each ABL, we then fit a logistic curve (using nlninfit in MATLAB) of the form

$$y = a + (b - a) \cdot \frac{1}{\exp(-c(x - d))}, \tag{3}$$

where $x$ is the ILD value and $y$ is the fraction of ipsilateral classifications (*Figure 6A–B*). Parameters $a$ and $b$ specify the left and the right asymptote values, parameter $c$ specifies the slope in the middle of the curve, and parameter $d$ specifies the horizontal position of the middle. A perfect classifier would have $a = 0$, $b = 1$, $c = \infty$, and $d = 0$. We defined just noticeable difference (JND) as the horizontal distance between the points on the curve that yield 75% accuracy ($y = .25$ and $y = .75$).

To have one single number to describe the classifier performance for each ABL, we used the integral performance for ILDs from $-20$ to $+20$ under our logistic fit, that is

$$\frac{1}{40}\left[\int_0^{20} y\,dx + \int_{-20}^0 (1-y)\,dx\right]. \tag{4}$$

This measure was used for *Figure 6C–H*. For several sessions/ABLs, nlninfit returned a convergence error indicating a poor logistic fit; we excluded these cases from the analysis (0 for *Figure 6C–E*, 2 for *Figure 6F*, 2 for *Figure 6G* and 4 for *Figure 6H*).

For the subsampling and time-resolved analyses shown in *Figure 6—figure supplement 1*, we used an accuracy simply averaged over all ILDs (as opposed to the integral accuracy estimated via the logistic fit), because logistic fits were unreliable for small numbers of neurons or short time windows.

## Model selection for decoder performance summaries

The integral accuracies shown in *Figure 6C–E* can be fit with three separate regressions (six parameters): one for each ABL. Our regression model described in the main text only had four parameters and was preferred by both AIC and BIC. We only report BIC as the more stringent criterion. For simplicity, we used the same regression model for three variants of the classification analysis (*Figure 6F–H*).

## Acknowledgements

We thank Adrien Jouary for comments on our manuscript and Kenneth Harris for suggesting to use an unbiased method of estimation of signal correlations. This work was supported by the Fundação Bial (Fellowship 389/14) and the German Ministry of Education and Research (BMBF, FKZ 01GQ1601) (DK), an HFSP postdoctoral scholarship LT 000442/2012 (JP-V), a doctoral fellowship from the Fundação para a Ciência e a Tecnologia (MV), the Champalimaud Foundation (CKM and AR), the Simons Collaboration on the Global Brain 543009 and the NIH U01 NS094288 grants (CKM), and by a Marie Curie Career Integration Grant PCIG11-GA-2012–322339, the HFSP Young Investigator Award RGY0089 and the EU FP7 grant ICT-2011-9-600925 (NeuroSeeker) (AR).

## Additional information

### Funding

| Funder | Grant reference number | Author |
|---|---|---|
| Fundação Bial | 389/14 | Dmitry Kobak |
| Federal Ministry of Education and Research | FKZ 01GQ1601 | Dmitry Kobak |
| Human Frontier Science Program | Postdoctoral fellowship - LT 000442/2012 | Jose L Pardo-Vazquez |
| Fundação para a Ciência e a Tecnologia | | Mafalda Valente |
| Champalimaud Foundation | | Christian K Machens Alfonso Renart |
| Simons Foundation | Simons Collaboration on the Global Brain (SCGB): 543009 | Christian K Machens |
| National Institutes of Health | U01 NS094288 | Christian K Machens |
| European Union Seventh Framework Programme | Marie Curie Career Integration Grant - PCIG11-GA-2012-322339 | Alfonso Renart |
| Human Frontier Science Program | Young Investigator Award - RGY0089 | Alfonso Renart |
| European Union Seventh Framework Programme | ICT-2011-9-600925 | Alfonso Renart |

The funders had no role in study design, data collection and interpretation, or the decision to submit the work for publication.

### Author contributions

Dmitry Kobak, Writing—original draft, Writing—review and editing, Designed and performed the data analysis; Jose L Pardo-Vazquez, Conceptualization, Writing—review and editing, Performed experiments; Mafalda Valente, Performed experiments; Christian K Machens, Supervision, Writing—review and editing; Alfonso Renart, Conceptualization, Supervision, Writing—original draft, Writing—review and editing

### Author ORCIDs

Dmitry Kobak (iD) http://orcid.org/0000-0002-5639-7209
Jose L Pardo-Vazquez (iD) https://orcid.org/0000-0003-4623-2440
Mafalda Valente (iD) http://orcid.org/0000-0002-1824-0462
Christian K Machens (iD) http://orcid.org/0000-0003-1717-1562
Alfonso Renart (iD) http://orcid.org/0000-0001-7916-9930

### Ethics

Animal experimentation: All procedures were reviewed and approved by the Champalimaud Centre for the Unknown animal welfare committee and approved by the PortugueseDirecçaoGeral de Veterinaria (Ref. No. 0421/000/000/2019).

### Decision letter and Author response

Decision letter https://doi.org/10.7554/eLife.44526.021
Author response https://doi.org/10.7554/eLife.44526.022

## Additional files

### Supplementary files

• Transparent reporting form
DOI: https://doi.org/10.7554/eLife.44526.019

### Data availability

The full MATLAB code for the analysis is located at https://github.com/dkobak/a1geometry (copy archived at https://github.com/elifesciences-publications/a1geometry). We made the spike count data (spike counts for each neuron for each stimulus presentation from −50 ms to 150 ms in 50 ms bins) available in the same repository. This allows most of our figures to be reproduced. The complete dataset that was collected, including spike time data not analysed here, is available upon reasonable request to the corresponding author.

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
