## [Decision Letter]

Thank you for submitting your article "State-dependent geometry of population activity in rat auditory cortex" for consideration by *eLife*. Your article has been reviewed by three peer reviewers, and the evaluation has been overseen by a Reviewing Editor and Eve Marder as the Senior Editor. The following individuals involved in review of your submission have agreed to reveal their identity: Dan Goodman (Reviewer #1).

The reviewers have discussed the reviews with one another and the Reviewing Editor has drafted this decision to help you prepare a revised submission.

Summary:

Kobak et al. study the representation of sounds in primary auditory cortex (A1) as a function of their overall loudness (absolute binaural level or ABL) and of their azimuth (inter-aural level difference or ILD). Using silicon probe recordings of populations of neurons in urethane anesthetized rats, the authors ask how the encoding of ABL and ILD depends on the brain state. They report on two main discoveries. First, they find that the more activated the state, the weaker is the preference of the neurons towards contralateral and/or loud sounds. Second, they consider the geometry of the firing rate vector, and report that in the more active states the representations of ABL and ILD become almost orthogonal to each other and to the axis of overall activity level.

The paper is well written and addresses a major issue of interest to both theorists and experimentalists. The results will help guide the theory of neural coding, and provide a useful framework for future analyses of population activity. What makes the study most interesting is the state dependence reported by the authors, which can be seen in simpler measures and is nicely summarized with their geometric picture. Prior work suggests that neural codes themselves are state dependent, and the current manuscript is a significant new contribution to this line of research.

However, a few issues were identified that need to be addressed before the manuscript can be published.

Essential revisions:

1) On the *z*-score(Mean firing rate) formula, Equation 1: I think the "subtracting 40" from the ABL values needs some better justification. Why 40? Also, I'd like to see a short explanation, and certainly a reference, justifying why the interaction term (ABL-40)*ILD is the right one for capturing the effects of gain modulation. Most importantly, given that the gain modulation is not working quite as expected, I would really like to know what happens when you remove that interaction term from your formula. This will yield different fits for *β*_ILD_ and *β*_ABL_. If you re-do the analyses of Figure 3 using those fits, are the results qualitatively the same? Given that many of your insights depend on interpreting these coefficients, I'd like to make sure these results don't depend too much on the precise form you chose for Equation 1.

2) It appears that the geometry analyses were carried out using spike counts over the entire 150ms duration of the stimulus. However the PSTHs (Figure 2 and Figure 2—figure supplement 1) demonstrate that in the active state (or in up states) the late part of the response consists of reduction in firing rate relative to the baseline, which is not the case for the inactive state. It should be clarified whether this suppression of firing is the key to the orthogonality of the representations of ABL or ILD, or if it holds for the onset response as well. Also, is the 1st mode (PC) of the onset response (20-40ms) the same as the 1st mode of the late (70-150ms) portion of the response? Related: is it the case that the values in Figure 3—figure supplement 3A and B are (almost) identical whereas the values in Figure 3—figure supplement 3E and F are not? I believe providing this information is of major importance, e.g. it might suggest that in the active state the response has a second part, which is absent in the inactive state.

3) One of the main shortcomings of the study is that it was performed under anesthesia, hence there is always the question of to what extent the results apply to awake and behaving rodents. This would be even more worrying if it turns out that suppression of firing by the auditory stimulus is key to the presented results (point 3 above), as that would rely on the existence of a highly active state which seems rather unique to urethane. Indeed, McGinley et al. (McGinley et al., 2015) suggest that the best brain state for auditory detection (in awake and behaving rodents) is of moderate rather than high activation. At the very least this point should be included in the Discussion.

4) In the analyses of the signal and noise correlation matrices, you compute (a) mean correlation, (b) fraction of variance by first PCA component, and (c) the dimensionality. This last item is not as self-explanatory as the first two, and it's important to say a bit more in the main text about what exactly you mean by "dimensionality," and how you estimate it. In other words, please don't relegate the entire discussion of this to the Materials and methods, as it is an important point.

5) If I understand correctly the evoked response of every recorded neuron was used. In such studies it is typical to find a substantial percentage of neurons without statistically significant sensory responses. First, it is important to report the percentage of neurons with significant ABL/ILD tuning as a function of brain state. Second, what is the impact of those non-tuned neurons? Some of the analyses (e.g. Figure 3) would benefit if such neurons were excluded. In fact, one probably should only use neurons that show significant tuning to ABL or ILD (or their interaction), otherwise neurons that simply respond to an up-state triggered by the stimulus (Figure 2—figure supplement 1D) would be included as well.

---

## [Author Response]

Essential revisions:1) On the z-score(Mean firing rate) formula, Equation 1: I think the "subtracting 40" from the ABL values needs some better justification. Why 40? Also, I'd like to see a short explanation, and certainly a reference, justifying why the interaction term (ABL-40)*ILD is the right one for capturing the effects of gain modulation. Most importantly, given that the gain modulation is not working quite as expected, I would really like to know what happens when you remove that interaction term from your formula. This will yield different fits for β_ILD_ and β_ABL_. If you re-do the analyses of Figure 3 using those fits, are the results qualitatively the same? Given that many of your insights depend on interpreting these coefficients, I'd like to make sure these results don't depend too much on the precise form you chose for Equation 1.

Thanks for bringing this up. Indeed, we agree that this was not explained clearly enough in our text. Subtracting 40 centers ABL predictor (remember that possible ABL values are 20, 40, and 60). Centering the predictors makes all three terms orthogonal, so the model with interaction term and the model without interaction term produce identical *β*_ILD_ and *β*_ABL_. This is only true if ILD and ABL are both centered.

If we did not subtract 40 in the model *without* interaction term, then *β*_ILD_ and *β*_ABL_ would stay exactly the same, only the intercept would change. However, not subtracting 40 (or subtracting something else) in the model *with* interaction can drastically changes *β*_ILD_ because the interaction term would not be orthogonal to the ILD term anymore. Overall, we think that working with centered predictors in a linear model – a standard statistical approach related to e.g. ANOVA – makes the coefficients much easier to interpret (they capture the slope of a uni-variate linear regression). We have added an explanation of the motivation for centering the ABL predictor in the second paragraph of the subsection “Evoked activity of single neurons”.

Regarding the relationship between the interaction term and gain modulation, in our mind it represented the simplest way to model a multiplicative effect of ABL on ILD tuning (of course more complicated gain models are possible (e.g. slope of ILD changing quadratically with ABL, etc.), but we think this simple model provides a reasonable first approximation). Gain modulation is used in the literature with two slightly different meanings. It can refer to a change in the slope of the *f −I* curve of a neuron, typically in response to changes in the dynamic range of the inputs (as in, e.g., Dahmen et al., Neuron 2010), or to a multiplicative modulation of tuning curves with respect to an external parameter (as in, e.g., the effect of contrast on orientation tuning; Incidentally, these two notions are related: both contrast-invariant orientation tuning as well as modulation of input-output transfer functions has been explained through divisive normalization (e.g., Heeger et al., Vis. Neurosci., 9, 181 (1992); Carandini and Heeger, Nat. Rev. Neurosci., 13, 51 (2012)). It is in this second sense that we talk about gain modulation of the tuning to ILD by ABL. We are not aware of any previous studies using the parametrisation that we selected to study the gain modulation of ILD by ABL, so we are unable to provide a reference, but the change in slope of the tuning curves for azimuth as a function of sound intensity is evident in many studies (e.g., first column on the left of Figure 6 in Stecker et al., 2005 – now mentioned in the first paragraph of the subsection “State-dependence of ILD-tuning at different sound levels”). We speculate that using our parametrisation to fit data like the one shown in this figure would have led to results regarding the effect of ABL on ILD tuning that are similar to the ones we see in the inactive state. Using the reviewer’s wording, we find that gain modulation works ‘as expected’ in the inactive recordings, but its effect is heterogeneous across the population in the active state.

2) It appears that the geometry analyses were carried out using spike counts over the entire 150ms duration of the stimulus. However the PSTHs (Figure 2 and Figure 2—figure supplement 1) demonstrate that in the active state (or in up states) the late part of the response consists of reduction in firing rate relative to the baseline, which is not the case for the inactive state. It should be clarified whether this suppression of firing is the key to the orthogonality of the representations of ABL or ILD, or if it holds for the onset response as well. Also, is the 1st mode (PC) of the onset response (20-40ms) the same as the 1st mode of the late (70-150ms) portion of the response? Related: is it the case that the values in Figure 3—figure supplement 3A and B are (almost) identical whereas the values in Figure 3—figure supplement 3E and F are not? I believe providing this information is of major importance, e.g. it might suggest that in the active state the response has a second part, which is absent in the inactive state.

That is a great point. The reviewers are indeed correct that the time course of the evoked responses is state-dependent. Thus, as pointed out by the referees, it is important to understand the extent to which the strong state-dependence of the neural code is mainly attributable to the late response, or holds also during the onset response period. The reviewers’ comments led us to study this in more depth, and we have included additional analyses into the manuscript.

First, we repeated our “geometry” analysis (Figure 5) using only onset (0–50 ms) responses, and added the result as a new supplementary figure (Figure 6—figure supplement 2). Most of our observations were unchanged. In particular, the angle between ABL and ILD in the onset response was still much higher in the active state compared to the inactive state (Figure 6—figure supplement 2, left panel). Overall correlation with CV got more noisy though.

Second, we assessed quantitatively the difference in single cell tuning between early and late responses displayed in Figure 3—figure supplement 3. Briefly, for every neuron significantly tuned in both early and late periods, we measured’ differences in tuning’ as the distance *d* between its *β*_ILD_ q and *β*_ABL_ coordinates, i.e.,d=βILDonset-βILDlate²+βABLonset-βABLlate². This analysis revealed that the distance between the “early” and the “late” locations was very similar in both states: it was 0.036 *±* 0.025 in the active state (mean *±* SD for 117 neurons present in panels A and B) and 0.036 *±* 0.023 in the inactive state (mean *±* SD for 41 neurons present in panels E and F). Thus, tuning does not appear to change in time differently across states during evoked responses. We did find, however, a difference across states in the *fraction* of neurons with significant tuning during the late period relative to the onset period. In the active state, there are significantly more neurons tuned during the late than during the onset period (41.8% vs 23.1%; *p* = 0.00001, Fisher’s exact test), whereas in the inactive state the number of neurons tuned during the early and late periods was not noticeably different (32.2% vs 31.5%; *p* = 0.8, Fisher’s exact test). This was added to the caption of Figure 3—figure supplement 3 and the surrounding text.

Third, a time-resolved decoding analysis (Figure 6—figure supplement 1, already present in the first version of our manuscript) allows one to assess significant trends in decoding accuracy as a function of time during the evoked response across states. This figure shows that decoding accuracy from only the first *∼* 50 ms is clearly state dependent. In the inactive state, early decoding accuracy improves with ABL, whereas in the active recordings it is approximately ABL-invariant. The ABL-invariance of sound lateralization decoding in the active state is consistent with the approximate orthogonality between the ABL and ILD axis identified by our geometrical analysis. Interestingly, decoding accuracy continues to increase after the onset response in the active but not in the inactive experiments, suggesting that the late response during cortical activation (characterized by an approximate balance of excitatory and inhibitory responses) depends on ILD in a way that is not redundant with onset activity. This is consistent and likely related to the appearance, in the ACT recordings, of tuned neurons during the late response which were not tuned in the onset response (see previous paragraph). In the inactive experiments, in contrast, decoding accuracy saturates after the onset (and the number of tuned neurons is not different between early and late periods).

Overall, these series of analyses suggest that while late responses contribute to the large reorganization of the neural code across states, they are not sufficient to explain it. Indeed, across most of the measures we quantified, the first 50 ms behave similarly as the overall stimulus duration (150 ms). In addition to the new Figure 6—figure supplement 2 and a description of these analyses on the Results section, we have added a subsection ‘Time-course of stimulus representations during the evoked response’ in the Discussion to summarize our conclusions on this issue.

3) One of the main shortcomings of the study is that it was performed under anesthesia, hence there is always the question of to what extent the results apply to awake and behaving rodents. This would be even more worrying if it turns out that suppression of firing by the auditory stimulus is key to the presented results (point 3 above), as that would rely on the existence of a highly active state which seems rather unique to urethane. Indeed, McGinley et al. (McGinley et al., 2015) suggest that the best brain state for auditory detection (in awake and behaving rodents) is of moderate rather than high activation. At the very least this point should be included in the Discussion.

We agree with the statement that the relationship between the type of desynchronization observed under urethane and the types observed during wakefulness is not fully understood yet. The desynchronized state under Urethane appears to be an accurate model of the naturally occurring brain state during R.E.M. sleep (Clement et al., 2008), but the relationship between cortical activation during R.E.M. and during wakefulness has, as far as we can tell, not been thoroughly characterized. In fact, the behavioral and cognitive determinants of different waking states is an area of active research. While in rodents there is substantial evidence that active exploration (running, whisking, etc) leads to strong desynchronization, the extent to which different cognitive tasks *without overt behavior* can fully desynchronize the rodent cortex is still an open issue. The fact that desynchronization under urethane and during selective attention both rely critically on cholinergic signalling suggests a possible link between these two types of desynchronization, but the issue is not easy to address, in part because it has proved difficult to design clean selective attention tasks in rodents. We have now touched on these issues in the’ Mechanisms of State Dependence’ subsection of the Discussion.

We fully agree that the McGinley et al. study is very relevant for our work, despite the fact that it addresses the role of brain state in sensory detection, i.e., for near-threshold stimuli, whereas we focus on supra-threshold intensities. We now discuss it in the’ Functional Implications’ subsection of the Discussion.

4) In the analyses of the signal and noise correlation matrices, you compute (a) mean correlation, (b) fraction of variance by first PCA component, and (c) the dimensionality. This last item is not as self-explanatory as the first two, and it's important to say a bit more in the main text about what exactly you mean by "dimensionality," and how you estimate it. In other words, please don't relegate the entire discussion of this to the Materials and methods, as it is an important point.

We agree with the reviewers and have now added a brief clarification about “dimensionality” in the fourth paragraph of the subsection “Signal and noise correlation matrices”. We have refrained from providing a lengthy explanation as this would likely make the results harder to follow, and we have kept the in-detail explanation of the permutation procedure in the Materials and methods.

5) If I understand correctly the evoked response of every recorded neuron was used. In such studies it is typical to find a substantial percentage of neurons without statistically significant sensory responses. First, it is important to report the percentage of neurons with significant ABL/ILD tuning as a function of brain state. Second, what is the impact of those non-tuned neurons? Some of the analyses (e.g. Figure 3) would benefit if such neurons were excluded. In fact, one probably should only use neurons that show significant tuning to ABL or ILD (or their interaction), otherwise neurons that simply respond to an up-state triggered by the stimulus (Figure 2—figure supplement 1D) would be included as well.

In both states around one half of the neurons showed significant linear tuning with overall *p <* 0.01 (53.5%, 489/914, in the inactive sessions and 49.5%, 154/311, in the active sessions). We added this information to the Results (subsection “Evoked activity of single neurons”, second paragraph).

We modified our single neuron figures (Figure 3 and corresponding supplementary figures) to only show neurons with *p <* 0.01. We thank the reviewer for this suggestion because we think this made the figures clearer. Also, it increased the strengths of all three effects shown in Figure 3G–I. We kept a version of Figure 3 showing all neurons as a supplementary figure for consistency with other analyses which are based on the population as a whole.